# Towards Hierarchical Rectified Flow

**Yichi Zhang[1], Yici Yan[1], Alex Schwing[1], Zhizhen Zhao[1]**
[1]University of Illinois Urbana-Champaign

## Abstract

We formulate a hierarchical rectified flow to model data distributions. It hierarchically couples multiple ordinary differential equations (ODEs) and defines a time-differentiable stochastic process that generates a data distribution from a known source distribution. Each ODE resembles the ODE that is solved in a classic rectified flow, but differs in its domain, i.e., location, velocity, acceleration, etc. Unlike the classic rectified flow formulation, which formulates a single ODE in the location domain and only captures the expected velocity field (sufficient to capture a multi-modal data distribution), the hierarchical rectified flow formulation models the multi-modal random velocity field, acceleration field, etc., in their entirety. This more faithful modeling of the random velocity field enables integration paths to intersect when the underlying ODE is solved during data generation. Intersecting paths in turn lead to integration trajectories that are more straight than those obtained in the classic rectified flow formulation, where integration paths cannot intersect. This leads to modeling of data distributions with fewer neural function evaluations. We empirically verify this on synthetic 1D and 2D data as well as MNIST, CIFAR-10, and ImageNet-32 data. Our code is available at: https://riccizz.github.io/HRF/.

## 1 Introduction

Diffusion models (Ho et al., 2020; Song et al., 2021a;b) and particularly also flow matching (Liu et al., 2023; Lipman et al., 2023; Albergo & Vanden-Eijnden, 2023; Albergo et al., 2023) have gained significant attention recently. This is partly due to impressive results that have been reported across domains from computer vision (Ho et al., 2020) and medical imaging (Song et al., 2022) to robotics (Kapelyukh et al., 2023) and computational biology (Guo et al., 2024). Beyond impressive results, flow matching was also reported to faithfully model multimodal data distributions. In addition, sampling is reasonably straightforward: it requires to solve an ordinary differential equation (ODE) via forward integration of a set of source distribution points along an estimated velocity field from time zero to time one. The source distribution points are sampled from a simple and known source distribution, e.g., a standard Gaussian.

The velocity field is obtained by matching velocities from a constructed "ground-truth" integration path with a parametric deep net using a mean squared error (MSE) objective. See Fig. 1(a) for the "ground-truth" integration paths of classic rectified flow. Studying the "ground-truth" velocity distribution at a distinct location and time for rectified flow reveals a multimodal distribution. We derive an analytic expression for the multimodal velocity distribution in case of a mixture-of-Gaussian data distribution in Section 3.1. It is known that the MSE objective used in classic rectified flow does not permit to capture this multimodal distribution. Instead, classic rectified flow leads to a model that aims to capture the mean of the velocity distribution. This is illustrated in Fig. 1(b).

We do want to emphasize that capturing the mean of the velocity distribution is sufficient for characterizing a multimodal data distribution (Liu et al., 2023). However, only capturing the mean velocity also leads to unnecessarily curved forward integration paths. This is due to the fact that integration paths cannot intersect when using an MSE objective, as can be observed in Fig. 1(b).

In this paper, we hence wonder whether it is possible to capture the velocity distribution in its entirety. This enables integration paths to intersect during data generation, as illustrated in Fig. 1(c). Intuitively, and as detailed in Section 3.2, we can capture the velocity distribution by formulating a rectified flow objective in the velocity space rather than the location space. Hence, instead of training

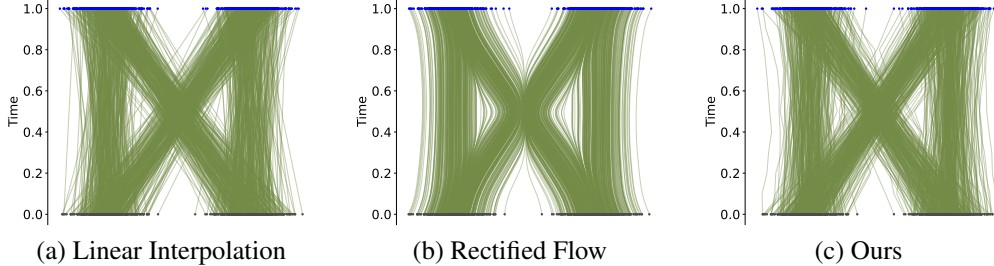

Figure 1: Particles flow from starting points (grey) to endpoints (blue) as time increases from 0 to 1. Ideally, the trajectories (green) are straight lines connecting two ends as shown in (a). Rectified Flow captures the expected velocity field while our Hierarchical Rectified Flow can model the true velocity field thus generating intersecting and more straight paths.

a deep net to estimate the velocity for integration in location space, as done in classic rectified flow, we train a deep net that estimates the acceleration for integration in velocity space. Sampling can then be done by forward integrating two hierarchically coupled processes: first, forward integrate in velocity space to obtain a sample from the velocity distribution; then use the velocity sample to perform a step in location space. While this nested integration of two processes seems computationally more demanding at first, it turns out that fewer integration steps are needed, particularly in the latter process. This is due to the fact that the integration path is indeed less curved, as shown in Fig. 1(c). We also show in Section 3.3 that capturing the velocity distribution in its entirety permits to capture a multimodal data distribution. The data generation process is governed by a random differential equation (RDE) (Han, 2018) with learned random velocity field.

Going forward, instead of using 'just' two hierarchically coupled processes we can extend the formulation to an arbitrary depth, which is detailed in Section 3.4. Using a depth of one defaults to classic rectified flow (deep net captures the expected velocity field), while a depth of two leads to a deep net that captures the acceleration, etc. We refer to this construction of hierarchically coupled processes as a 'hierarchical rectified flow.'

Empirically, we find that the studied hierarchical rectified flow leads to samples that better fit the data distribution. Specifically, we find that this hierarchical rectified flow leads to slightly better results than the vanilla rectified flow.

## 2 PRELIMINARIES

Given a dataset $\mathcal{D} = \{(x_1)\}$ consisting of samples $x_1 \sim \rho_1$, e.g., images, drawn from an unknown target data distribution $\rho_1$, the goal of generative modeling is to learn a model that faithfully captures the unknown target data distribution $\rho_1$ and permits to sample from the learned distribution.

Since we focus primarily on rectified flow, we provide its formulation in the following. At inference time, rectified flow starts from samples $x_0 \sim \rho_0$ drawn from a known source distribution $\rho_0$, e.g., a standard Gaussian. The source distribution samples are pushed forward from time $t = 0$ to target time $t = 1$ via integration along a trajectory specified via a learned velocity field $v(z_t, t)$. This learned velocity field depends on the current time $t$ and the sample location $z_t$ at time $t$. Formally, we obtain samples by numerically solving the ordinary differential equation (ODE)

$$dz_t = v(z_t, t)dt, \text{ with } z_0 \sim \rho_0, \quad t \in [0, 1]. \tag{1}$$

Notably, this sampling procedure is able to capture multimodal dataset distributions, as one expects from a generative model.

To learn the velocity field, at training time, rectified flow constructs random pairs $(x_0, x_1)$, consisting of a source distribution sample $x_0 \sim \rho_0$ and a target distribution sample $x_1 \sim \mathcal{D}$. The latter is drawn from a given dataset $\mathcal{D}$ consisting of samples which are assumed to be drawn from the unknown target distribution $\rho_1$. For a uniformly drawn time $t \sim U[0, 1]$, the time-dependent location $x_t$ is computed from the pair $(x_0, x_1)$ using linear interpolation of $(x_0, x_1)$, i.e.,

$$x_t = (1 - t)x_0 + tx_1, \quad \text{where } x_0 \sim \rho_0, \ x_1 \sim \mathcal{D}. \tag{2}$$

At this location $x_t$ and time $t$, the "ground-truth" velocity $v_{\text{gt}}(x_t, t) = \partial x_t / \partial t = x_1 - x_0$ is readily available. It is then matched during training with a velocity model $v(x_t, t)$ via a standard $\ell_2$ loss, i.e., during training we address

$$\inf_v \mathbb{E}_{x_0 \sim \rho_0, x_1 \sim \mathcal{D}, t \sim U[0,1]} \left[ \|x_1 - x_0 - v(x_t, t)\|_2^2 \right], \tag{3}$$

where the optimization is over the set of all measurable velocity fields. In practice, the functional velocity model $v(x_t, t)$ is often parameterized via a deep net with trainable parameters $\theta$, i.e., $v(x_t, t) \approx v_\theta(x_t, t)$, and the infimum resorts to a minimization over parameters $\theta$.

Considering the training procedure more carefully, it is easy to see that different random pairs $(x_0, x_1)$ can lead to different "ground-truth" velocity directions at the same time $t$ and at the same location $x_t$. The aforementioned $\ell_2$ loss hence asks the functional velocity model $v(x_t, t)$ to regress to different "ground-truth" velocity directions. This leads to averaging, i.e., the optimal functional velocity model $v^*(x_t, t) = \mathbb{E}_{\{(x_0, x_1, t):(1-t)x_0 + tx_1 = x_t\}} [v(x_t, t)]$.

According to Theorem 3.3 by Liu et al. (2023), if we use $v^*$ for the ODE in Eq. (1), then the stochastic process associated with Eq. (1) has the same marginal distributions for all $t \in [0, 1]$ as the stochastic process associated with the linear interpolation characterized in Eq. (2).

Nonetheless, to avoid the averaging, in this paper we wonder whether it is possible to capture the multimodal velocity distribution at each time $t$ and at each location $x_t$, and whether there are any potential benefits to doing so.

# 3 TOWARDS HIERARCHICAL RECTIFIED FLOW

In the following Section 3.1, we first discuss the multimodality of the velocity distribution and provide a case study with Gaussian mixtures. The case study is designed to provide insights regarding the velocity distribution. We then discuss in Section 3.2 a simple way to capture the multimodal velocity distribution and how to use it to sample from the data distribution. Then, we show in Section 3.3 that the proposed procedure indeed faithfully captures the data marginals. Finally, we discuss in Section 3.4 an extension towards a hierarchical rectified flow formulation.

## 3.1 VELOCITY DISTRIBUTION AND CASE STUDY WITH GAUSSIAN MIXTURES

The linear interpolation in Eq. (2) defines a time-differentiable stochastic process with the random velocity field $v(x_t, t) = x_1 - x_0$, where $x_0 \sim \rho_0$ and $x_1 \sim \rho_1$. Note, the source and target distributions are independent. The following theorem characterizes the distribution of the velocity at a specific space time location $(x_t, t)$:

**Theorem 1** *The velocity distribution $\pi_1(v; x_t, t)$ at the space time location $(x_t, t)$ induced by the linear interpolation in Eq. (2) is*

$$\pi_1(v; x_t, t) = p_{V|X_t}(v|x_t) = \frac{\rho_0(x_t - tv)\rho_1(x_t + (1-t)v)}{\rho_t(x_t)}, \tag{4}$$

*for $\rho_t(x_t) \neq 0$ with ('*' denotes convolution)*

$$\rho_t(x_t) = \begin{cases} \rho_0(x_0) & \text{for } t = 0, \\ \frac{1}{t(1-t)} \rho_0 \left( \frac{x_t}{1-t} \right) * \rho_1 \left( \frac{x_t}{t} \right) & \text{for } t \in (0, 1), \\ \rho_1(x_1) & \text{for } t = 1. \end{cases} \tag{5}$$

*The distribution $\pi_1(v; x_t, t)$ is undefined if $\rho_t(x_t) = 0$.*

The proof of Theorem 1 is deferred to Appendix A. Note that since $\rho_1$ is typically multimodal, the resulting $\pi_1(v; x_t, t)$ is also multimodal. At $t = 0$, we have $\pi_1(v; x_t, t) = \rho_1(x_t + v)$, which corresponds to the data distribution shifted by $-x_t$. At $t = 1$, we have $\pi_1(v; x_t, t) = \rho_0(x_t - v)$, which corresponds to the flipped source distribution shifted by $x_t$.

To illustrate the multimodality of the velocity distribution, we consider a simple 1-dimensional example. The source distribution is a standard Gaussian (zero mean, unit variance). The target distribution is a Gaussian mixture. The following corollary provides the "ground-truth" velocity distribution at any location $x_t$.

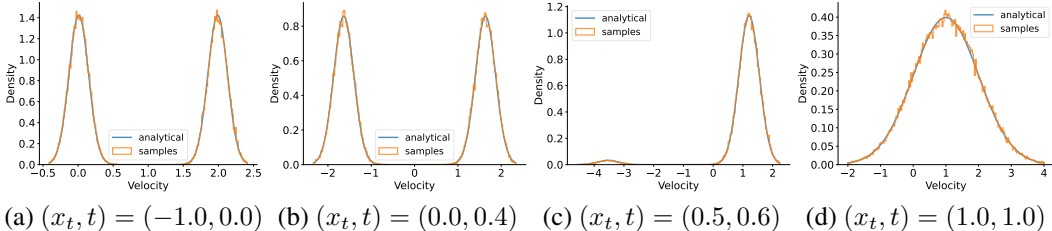

(a) $(x_t, t) = (-1.0, 0.0)$ (b) $(x_t, t) = (0.0, 0.4)$ (c) $(x_t, t) = (0.5, 0.6)$ (d) $(x_t, t) = (1.0, 1.0)$

Figure 2: We verify the derived velocity distribution by comparing its probability density function (blue) to the empirical sample histogram (orange) at different times $t$ and locations $x_t$.

**Corollary 1** *Assume $\rho_0 = \mathcal{N}(x; 0, 1)$ and $\rho_1 = \sum_{k=1}^{K} w_k \mathcal{N}(x; \mu_k, \sigma_k^2)$, then*

$$\pi_1(v; x_t, t) = \sum_{k=1}^{K} \tilde{w}_{k,t} \mathcal{N}\left(v; \frac{(1-t)(\mu_k - x_t) + t\sigma_k^2 x_t}{\tilde{\sigma}_{k,t}^2}, \frac{\sigma_k^2}{\tilde{\sigma}_{k,t}^2}\right), \tag{6}$$

*where $\tilde{\sigma}_{k,t}^2 = (1-t)^2 + t^2 \sigma_k^2$ and $\tilde{w}_{k,t} = \frac{w_k \mathcal{N}(x_t; t\mu_k, \tilde{\sigma}_{k,t}^2)}{\sum_{k'=1}^{K} w_{k'} \mathcal{N}(x_t; t\mu_{k'}, \tilde{\sigma}_{k',t}^2)}$.*

We defer the proof of Corollary 1 to Appendix B. To empirically check the fit of Corollary 1, in Fig. 2, we compare the derived velocity distribution with empirical estimates at different locations $(x_t, t)$. We observe a great fit and very clearly multimodal distributions.

It is very much worthwhile to study these distributions a bit more. In particular, we observe that the velocity distribution at time $t = 1$ collapses to a single Gaussian, more specifically a shifted source distribution. This can be seen from Fig. 2(d). Further, at time $t = 0$, we observe the velocity distribution to be identical to a shifted data distribution. This can be seen from Fig. 2 (a).

This is valuable to know as it suggests that the velocity distribution is at least as complex as the data distribution. Indeed, at $t = 0$, the velocity distribution is identical to a shifted data distribution.

## 3.2 MODELING THE VELOCITY DISTRIBUTION

The previous section showed that the velocity distributions can be multimodal. Knowing that the optimal velocity model $v^*(x_t, t)$ of classic rectified flow averages "ground-truth" velocities, we can't expect classic rectified flow to capture this distribution. We hence wonder: 1) is it possible to capture the multimodal velocity distribution at each time $t$ and at each location $x_t$; 2) are there any benefits to capturing the multimodal velocity distribution as opposed to 'just' capturing its mean as done by classic rectified flow.

Intuitively, an accurate characterization of the velocity distribution might be beneficial because we obtain straighter integration paths, which in turn may lead to easier integration with fewer neural function evaluations (NFE). In addition, capturing the velocity distribution provides additional modeling flexibility (an additional time axis), which might yield to improved results. Notably, modeling of the velocity distribution does not lead to modeling of a simpler distribution. As mentioned in Section 3.1, at time $t = 0$ the velocity distribution is identical to a shifted data distribution.

To accurately model the "ground-truth" velocity distribution, we can use rectified flow for velocities rather than locations, which are used in the classic rectified flow formulation. This is equivalent to learning the acceleration. To see this, first, consider classic rectified flow again: we construct a time-dependent location $x_t$ from pairs $(x_0, x_1)$, compute the "ground-truth" velocity $v_{gt}(x_t, t) = \partial x_t / \partial t$, and train a velocity model $v_\theta(x_t, t)$ to match this "ground-truth" velocity $v_{gt}(x_t, t)$.

To learn the acceleration, we introduce a source velocity sample $v_0 \sim \pi_0$ drawn from a known source velocity distribution $\pi_0$. We also construct a target velocity sample $v_1(x_t, t) \sim \pi_1(v; x_t, t)$ at time $t$ and at location $x_t$, which follows the target velocity distribution $\pi_1(v; x_t, t)$ at time $t$ and at location $x_t$. Note, the target velocity sample at time $t$ and at location $x_t = (1-t)x_0 + tx_1$ is obtained via $v_1(x_t, t) = x_1 - x_0$, when considering a rectified flow. The samples $v_1(x_t, t)$ follow the "ground-truth" velocity distribution $\pi_1(v; x_t, t)$ at time $t$ and at location $x_t$.

---

**Algorithm 1:** Hierarchical Rectified Flow Training

---

1 The source distributions $\rho_0$ and $\pi_0$ and the dataset $\mathcal{D}$
2 **while** *stopping conditions not satisfied* **do**
3     Sample $x_0 \sim \rho_0, x_1 \sim \mathcal{D}$, and $v_0 \sim \pi_0$ ;          //better to sample a mini-batch
4     Sample $t \sim U[0,1]$ and $\tau \sim U[0,1]$ ;  //different $t$ and $\tau$ for each mini-batch sample
5     Compute loss following Eq. (8);
6     Perform gradient update on $\theta$
7 **end**

---

Using both the source velocity sample $v_0$ and the target velocity sample $v_1(x_t, t)$, and following classic rectified flow, we introduce a new time-axis $\tau \in [0,1]$ and construct a time-dependent velocity $v_\tau(x_t, t) = (1 - \tau)v_0 + \tau v_1(x_t, t)$ at time $t$ and at location $x_t$. Using it, we obtain the "ground-truth" acceleration from the time-dependent velocity $v_\tau(x_t, t)$ via $a_{\text{gt}}(x_t, t, v_\tau, \tau) = \partial v_\tau / \partial \tau = v_1(x_t, t) - v_0 = x_1 - x_0 - v_0$.

Note, for a specific $(x_t, t)$, we can get the following ODE induced from the linear interpolation of the target velocity distribution to convert $u_0 \sim \pi_0$ to $u_1 \sim \pi_1(v; x_t, t)$,

$$du_\tau(x_t, t) = a(x_t, t, u_\tau, \tau)d\tau, \quad \text{with } u_0 \sim \pi_0. \tag{7}$$

Here, $a(x_t, t, u_\tau, \tau) = \mathbb{E}_{\pi_0, \pi_1(v; x_t, t)}[V_1 - V_0 | V_\tau = u_\tau] = \mathbb{E}_{\pi_0, \rho_0, \rho_1}[X_1 - X_0 - V_0 | V_\tau = u_\tau, X_t = x_t]$ is the expected acceleration vector field.

Our approach aims to learn the acceleration vector field $a$ though flow matching for all $(x_t, t)$, i.e., matching the "ground-truth" acceleration by addressing

$$\inf_a \mathbb{E}_{x_0 \sim \rho_0, x_1 \sim \mathcal{D}, t \sim U[0,1], v_0 \sim \pi_0, \tau \sim U[0,1]} \left[ \|(x_1 - x_0 - v_0) - a(x_t, t, v_\tau, \tau)\|_2^2 \right]. \tag{8}$$

In practice, we use a parametric model $a_\theta(x_t, t, v_\tau, \tau)$ to match the target "ground-truth" acceleration by minimizing the objective w.r.t. the trainable parameters $\theta$. Training of the parametric acceleration model is straightforward. It is summarized in Algorithm 1.

It remains to answer how we use the trained acceleration model $a_\theta(x_t, t, v_\tau, \tau)$ during sampling. We have the following coupled ODEs induced from the coupled linear interpolations:

$$\begin{cases} du_\tau(z_t, t) = a(z_t, t, u_\tau, \tau)d\tau, & \text{with } u_0(z_t, t) \sim \pi_0, \quad \tau \in [0,1], \\ dz_t = u_1(z_t, t)dt, & \text{with } z_0 \sim \rho_0, \quad t \in [0,1]. \end{cases} \tag{9}$$

Those coupled ODEs convert $z_0 \in \rho_0$ to $z_1 \in \rho_1$. After training, the ODEs in Eq. (9) are simulated using the vanilla Euler method and $a_\theta$, as detailed in Algorithm 2. We first draw two random samples: $v_0 \sim \pi_0$ from the source velocity distribution and $x_0 \sim \rho_0$ from the source location distribution. We then integrate the velocity forward to time $\tau = 1$ to obtain a sample from the modeled velocity distribution $v_1(x_0, 0)$. Subsequently, we use this sample to perform one integration step on the location. We continue this procedure until we arrive at a sample $x_1$.

*Remark.* The generation of data is governed by a random differential equation (RDE) with the random velocity field, where Eq. (9) can be viewed as

$$dz_t = g(z_t, t, u_0(z_t, t))dt, \quad \text{where } u_0(z_t, t) \sim \pi_0, z_0 \sim \rho_0, \tag{10}$$

and $g$ is a deterministic function. The randomness comes from the initial conditions for data and velocity. This is different from the sampling process governed by a stochastic differential equation (SDE) used in diffusion models (Song et al., 2021b), where the randomness comes from the Wiener process and the initial condition for data.

Note that our use of the term acceleration is not due to second-order derivatives of the location, but rather due to two hierarchically coupled linear processes. We hence refer to this construction as a hierarchical rectified flow.

It remains to show that the obtained samples indeed follow the target data distribution. We will dive into this topic next.

---

**Algorithm 2:** Hierarchical Rectified Flow Sampling

---

**Input** : The source distributions $\rho_0$ and $\pi_0$, the number of $t$-discretization steps $J$, the number of $\tau$-discretization steps $L$, and the trained network parameters $\theta$.

1   Sample $z_0 \sim \rho_0$ and $u_0 \sim \pi_0$;
2   Compute $\Delta t = \frac{1}{J-1}$ and $\Delta \tau = \frac{1}{L-1}$;
3   **for** $j = 1, \ldots, J$ **do**
4     **for** $l = 1, \ldots, L$ **do**
5       Compute $u_l = u_{l-1} + a_\theta(z_{t_{j-1}}, t_{j-1}, u_{l-1}, \tau_{l-1}) \cdot \Delta \tau$
6     **end**
7     Compute $z_j = z_{j-1} + u_L \cdot \Delta t$
8   **end**

---

### 3.3   Discussions on the generated data distribution

We discuss below the property of the hierarchical rectified flow defined in Eq. (9). According to rectified flow theory, we can generate samples from the velocity distribution using the expected acceleration field. The following theorem states that the generation process defined in Eq. (9), which uses the velocity distribution, leads to correct marginals for all times $t \in [0, 1]$.

**Theorem 2** *The time-differentiable stochastic process $\boldsymbol{Z} = \{Z_t : t \in [0, 1]\}$ generated by Eq. (9) has the same marginal distribution as the time-differentiable stochastic process $\boldsymbol{X} = \{X_t : t \in [0, 1]\}$ generated by the linear interpolation in Eq. (2).*

We defer the proof of Theorem 2 to Appendix C. Intuitively, the marginal preserving property is because at each time $t \in [0, 1]$, we can express $z_t$ as the linear interpolation of an $x_0 \sim \rho_0$ and an $x_1 \sim \rho_1$ according to Eq. (2).

A key benefit of our approach is that the process $\boldsymbol{Z}$ can be piece-wise straight. Starting with samples $z_t$ from $\rho_t$ for $t \in [0, 1]$, we propagate each sample by $v(z_t, t)\Delta t$, where $v(z_t, t) \sim \pi_1(v; z_t, t)$. Since $v(z_t, t) = x_1 - x_0$, where $tx_1 + (1 - t)x_0 = z_t$, the straight path following $v(z_t, t)$ will lead to a sample from the data distribution. In other words, $\Delta t$ can be chosen arbitrarily in the interval $(0, 1 - t]$. In practice, the learned velocity distribution is not perfect. Therefore, instead of one-step generation from the initial distribution, we choose to propagate the samples for a couple of steps. As shown in Section 4, we typically only use 2-5 steps in the numerical integration for data generation. Computationally, straight paths are very attractive as trajectories with nearly straight paths incur small time-discretization error in numerical simulation.

### 3.4   Extending Towards Hierarchical Rectified Flow

Consider the training objective for acceleration matching discussed in Eq. (8), and further consider the coupled ODE solved when sampling from the constructed process as specified in Eq. (9). It is straightforward to extend both to an arbitrary depth. I.e., instead of modeling the velocity distribution by matching accelerations, we can model the acceleration distribution by matching jerk or go even deeper towards snap, crackle, pop, and beyond.

Formally, the training objective of a hierarchical rectified flow of depth $D$ is given by

$$\inf_f \mathbb{E}_{\boldsymbol{x}_0 \sim \boldsymbol{\rho}_0, x_1 \sim \rho_1, \boldsymbol{t} \sim U[0,1]^D} \left[ \left\| (x_1 - \mathbf{1}_D^T \boldsymbol{x}_0) - f(\boldsymbol{x_t}, \boldsymbol{t}) \right\|_2^2 \right]. \tag{11}$$

Here, $\mathbf{1}_D$ is the $D$-dimensional all-ones vector and $\boldsymbol{t} = \left[ t^{(1)}, \ldots, t^{(D)} \right]^T$ is a $D$-dimensional vector of time variables drawn from a $D$-dimensional unit cube $U[0, 1]^D$. Moreover, we use the $D$-dimensional vector of source distribution samples $\boldsymbol{x}_0 = [x_0^{(1)}, \ldots, x_0^{(D)}]^T$, drawn from a $D$-dimensional source distribution $\boldsymbol{\rho}_0$, e.g., a $D$-dimensional standard Gaussian. We further use the $D$-dimensional location vector $\boldsymbol{x_t} = [x_t^{(1)}, \ldots, x_t^{(D)}]^T$, with its $d$-th entry given as $x_t^{(d)} = (1 - t^{(d)})x_0^{(d)} + t^{(d)}(x_1 - \sum_{k=1}^{d-1} x_0^{(k)})$. In addition, we refer to $f$ as the functional field of directions. Note that Eq. (11) is identical to Eq. (3) if $D = 1$ or Eq. (8) if $D = 2$.

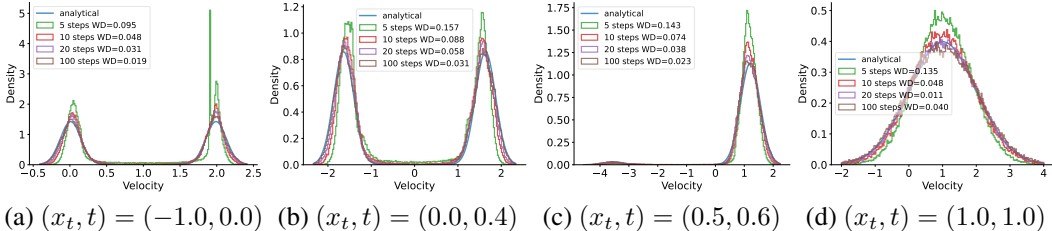

(a) $(x_t, t) = (-1.0, 0.0)$    (b) $(x_t, t) = (0.0, 0.4)$    (c) $(x_t, t) = (0.5, 0.6)$    (d) $(x_t, t) = (1.0, 1.0)$

Figure 3: Numerical estimation of $\pi_1(x_t, t)$ in HRF2 with different number of $v$ integration steps. The blue line shows the ground-truth $\pi_1$, where $\rho_0$ is a standard Gaussian and $\rho_1$ is a mixture of two Gaussians. The 1-Wasserstein distances (WD) for the estimates w.r.t. $\pi_1$ are shown in the legend.

Before discussing inference we want to highlight the importance of the first term in Eq. (11). Subtracting a large number of Gaussians from a data sample $x_1$ leads to a smoothed distribution. This is another potential benefit of a hierarchical rectified flow formulation.

Given a trained functional field of directions $f$ we sample from the defined process via numerical simulation according to the following coupled ODEs:

$$
\begin{cases}
dz_t^{(D)}\left(z_t^{(1:D-1)}, t^{(1:D-1)}\right) = f(z_t, t)dt^{(D)}, & \text{with } z_0^{(D)} \sim \rho_0^{(D)}, \\
dz_t^{(D-1)}\left(z_t^{(1:D-2)}, t^{(1:D-2)}\right) = z_1^{(D)}\left(z_t^{(1:D-1)}, t^{(1:D-1)}\right) dt^{(D-1)}, & \text{with } z_0^{(D-1)} \sim \rho_0^{(D-1)}, \\
\quad \vdots \\
dz_t^{(1)} = z_1^{(2)}\left(z_t^{(1)}, t^{(1)}\right) dt^{(1)}, & \text{with } z_0^{(1)} \sim \rho_0^{(1)}.
\end{cases}
\tag{12}
$$

Note that Eq. (12) is identical to Eq. (1) if $D = 1$ or Eq. (9) if $D = 2$.

Again, note that our use of the terms acceleration, jerk, etc. is not due to second, third, and higher-order derivatives of the location, but rather due to hierarchically coupled linear processes.

## 4 EXPERIMENTS

The studied hierarchical rectified flow (HRF) formulation couples multiple ODEs to accurately model the multimodal velocity distribution. To assess efficacy of this formulation, we first validate the approach in low-dimensional settings, where the analytical form of the velocity distribution is straightforward to compute. This allows us to verify that the model can indeed capture the velocity distribution accurately. We then investigate whether fitting the velocity distribution enhances the model's ability to fit the data distribution in generative tasks. We perform experiments on 1D data (Section 4.1), 2D data (Section 4.2), and high-dimensional image data (Section 4.3) with depth two HRF (HRF2) models: the models not only fit the velocity distribution but also enhance the quality of the generative process. We also include results for depth three HRF (HRF3) models on low dimensional data to show the potential for exploring deeper hierarchical structures. Importantly, for all experiments we report *total number of function evaluations (NFEs)*, i.e., the product of the number of integration steps at all HRF levels.

### 4.1 SYNTHETIC 1D DATA

For the 1D experiments, we first consider a standard Gaussian source distribution and a target distribution represented by a mixture of two Gaussians. Using Eq. (6), we can compute the analytical form of the velocity distribution. As shown in Fig. 3, our model captures the analytic velocity distribution with high accuracy. As expected, an increasing number of velocity integration steps increases the accuracy of the estimated velocity. The only exception occurs when $t$ approaches 1. The model performance deteriorates, and excessive steps accumulate errors, leading to less accurate results.

Next, we examine the data generation quality. We use a mixture of five Gaussians as illustrated in Fig. 4(a) and compare HRF to a baseline rectified flow (RF). We use the 1-Wasserstein distance (WD) as a metric to assess the quality of the generated data. As shown in Fig. 4(b), for the same

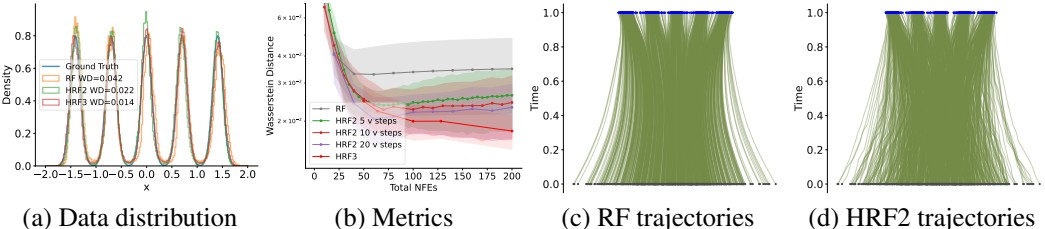

(a) Data distribution    (b) Metrics    (c) RF trajectories    (d) HRF2 trajectories

Figure 4: Results on 1D example, where $\rho_0$ is a standard Gaussian and $\rho_1$ is a mixture of 5 Gaussians. (a) Histograms of generated samples and $\rho_1$. (b) The 1-Wasserstein distance vs. NFE. (c) and (d) The trajectories of particles flowing from source distribution (grey) to target distribution (blue).

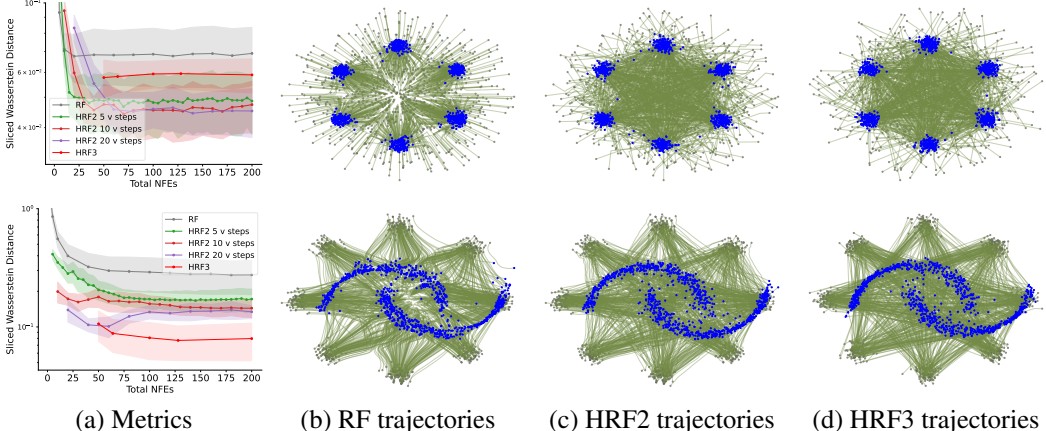

(a) Metrics    (b) RF trajectories    (c) HRF2 trajectories    (d) HRF3 trajectories

Figure 5: Results on 2D data. Top row: $\rho_0$ is a standard Gaussian and $\rho_1$ is a mixture of 6 Gaussians. Bottom row: $\rho_0$ is a mixture 8 Gaussians and $\rho_1$ is represented by the moons data. (a) Sliced 2-Wasserstein distance with respect to NFE. (b) and (c) show the trajectories (green) of sample particles flowing from source distribution (grey) to target distribution (blue).

neural function evaluations (NFEs), the HRF models outperform the baseline, producing data distributions with a lower WD, indicating superior quality. In Fig. 4's legend, the term "v steps" refers to the number of velocity integration steps. In this 1D experiment, HRF3 demonstrates better performance compared to HRF2. More 1D results are provided in Appendix H.1.

Additionally, we observe a fundamental difference in the generated trajectories. Since rectified flow estimates only the mean of the velocity distribution, it tends to move towards the center of the target distributions initially. In contrast, the HRF model determines the next direction at each space-time location based on the current velocity distribution. As shown in Fig. 4(d), the HRF2 trajectories are nearly linear and can intersect, which permits to use fewer data sampling steps during generation.

For the deep net, we use simple embedding layers and linear layers to first process the space and time information separately. Afterward, we concatenate these representations. This combined input is then passed through a series of fully connected layers, allowing the model to capture complex interactions and extract high-level features essential for accurate velocity prediction. We use the same architecture for the baseline model but increase the dimension of the hidden layers to optimize its performance. In contrast, the HRF2 model contains only 74,497 parameters compared to 297,089 parameters for the baseline model. This demonstrates the potential efficiency of HRF in handling higher-dimensional data while maintaining a more compact architecture. More details of the experiments are provided in Appendix F.

## 4.2 SYNTHETIC 2D DATA

For the 2D experiments, we consider two settings: 1) a standard Gaussian source distribution and a target distribution consisting of a mixture of six Gaussians; and 2) a mixture of eight Gaussians as the source distribution and the moons dataset as the target distribution. We employ the same network architecture as used in the 1D experiments. Due to the 2D data, we now have 76,674 parameters for

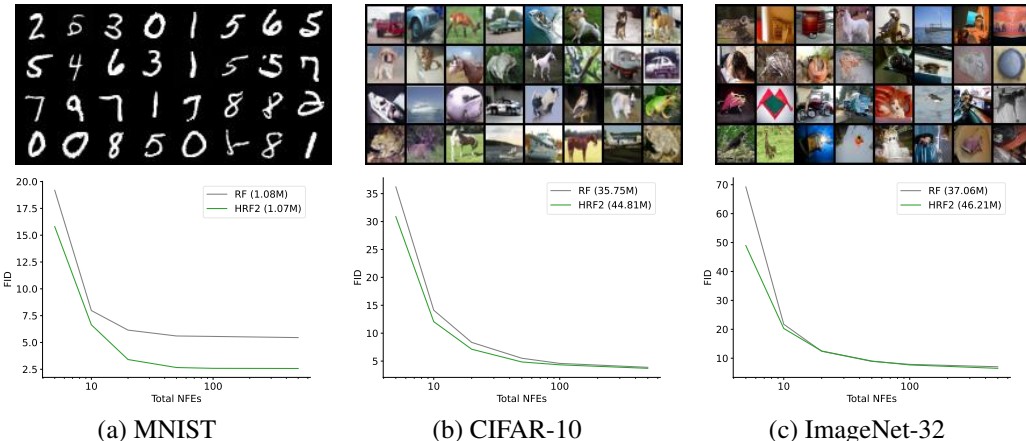

(a) MNIST        (b) CIFAR-10        (c) ImageNet-32

Figure 6: Experimental results on (a) MNIST, (b) CIFAR-10, and (c) ImageNet-32 datasets. Top row: samples of generated images, bottom row: FID scores with respect to total NFEs.

the HRF2 model and 329,986 parameters for the baseline RF model. We measure the quality of data generation using the sliced 2-Wasserstein distance (SWD). Fig. 5 shows the results. It is evident that on these more complex datasets, the performance gap between an HRF model and the rectified flow baseline is more pronounced. The trajectories demonstrate similar patterns to those observed in the 1D experiments: HRF2 produces significantly straighter paths, while the rectified flow baseline often exhibits large directional changes. Additionally, the HRF models consistently achieve higher quality in data generation compared to the baseline. HRF3 outperforms HRF2 for generating the moon data from a mixture of 8 Gaussians. However, HRF2 works better for the simpler mixture of Gaussian target. There is room to improve the training and scheduling of the integration steps among different layers for deeper HRF models.

### 4.3 IMAGE DATA

In addition to low-dimensional data, we also conduct experiments on high-dimensional image datasets including MNIST (LeCun et al., 1998), CIFAR-10 (Krizhevsky, 2009), and ImageNet-32 (Deng et al., 2009). We employ the Fréchet Inception Distance (FID) as the metric for evaluating image generation quality. For the baseline, we use the same U-Net architecture as Lipman et al. (2023) and successfully reproduced the state-of-the-art results across all datasets. The HRF model builds upon this U-Net structure. We follow the parameter settings and training procedures from Tong et al. (2024) and Lipman et al. (2023). Further details on the architecture and training setup are provided in Appendix F.

As shown in Fig. 6, for the same total NFEs, the HRF2 model demonstrates better performance on MNIST and CIFAR-10, and on-par performance on ImageNet-32 when compared to the baseline.

### 5 RELATED WORK

**Generative Modeling:** GANs (Goodfellow et al., 2014; Arjovsky et al., 2017), VAEs (Kingma & Welling, 2014), and normalizing flows (Tabak & Turner, 2013; Rezende & Mohamed, 2015; Dinh et al., 2017; Huang et al., 2018; Durkan et al., 2019) are classic methods for learning deep generative models. GANs excel in generating high-quality images but face challenges like training instability and mode collapse due to their min-max update mechanism. VAEs and normalizing flows rely on maximum likelihood estimation (MLE) for training, which necessitates architectural constraints or special approximations to ensure manageable likelihood computations. VAEs often employ a conditional Gaussian distribution alongside variational approximations, while the discrete normalizing flows utilize specifically designed invertible architectures and require costly Jacobian matrix calculations. Extending the discrete normalizing flow to continuous cases enabled the Jacobian to be unstructured yet estimatable using trace estimation methods (Hutchinson, 1990; Chen et al., 2018; Grathwohl et al., 2019). However, using maximum likelihood estimation (MLE) for this mapping

requires costly backpropagation through numerical integration. Regularizing the path can minimize solver calls (Finlay et al., 2020; Onken et al., 2021), but it doesn't resolve the fundamental optimization challenges. Rozen et al. (2021); Ben-Hamu et al. (2022) considered simulation-free training by fitting a velocity field, but still present scalability issues (Rozen et al., 2021) and biased optimization (Ben-Hamu et al., 2022).

Recent research has utilized diffusion processes, particularly the Ornstein-Uhlenbeck (OU) process, to link the target distribution $\rho_1$ with a source distribution $\rho_0$. This involves a stochastic differential equation (SDE) that evolves over infinite time, framing generative model learning as fitting the reverse evolution of the SDE from Gaussian noise to $\rho_1$ (Sohl-Dickstein et al., 2015; Ho et al., 2020; Song et al., 2021b). This method learns the velocity field by estimating the score function $\nabla \log(\rho_t(x))$ using the Fischer divergence instead of the maximum likelihood estimation (MLE) objective. Although diffusion models have demonstrated significant potential for modeling high-dimensional distributions (Rombach et al., 2022; Hoogeboom et al., 2022; Saharia et al., 2022), the requirement for infinite time evolution, heuristic time step parameterization (Xiao et al., 2022), and the unclear significance of noise and score (Bansal et al., 2024; Lu et al., 2022) pose challenges. Notably, the score-based diffusion models typically require a large number of time steps to generate data samples. In addition, calculating the actual likelihoods necessitates using the ODE probability flow linked to the SDE (Song et al., 2021b). These highlight the need for further exploration of effective ODE-driven methods for learning the data distribution.

**Flow Matching:** Concurrently, Liu et al. (2023); Lipman et al. (2023); Albergo & Vanden-Eijnden (2023) presented an alternative to score-based diffusion models by learning the ODE velocity through a time-differentiable stochastic process defined by interpolating between samples from the source and data distributions, i.e., $x_t = \psi_t(x_0, x_1)$, with $x_0 \sim \rho_0$ and $x_1 \sim \rho_1$, instead of the OU process. This offers greater simplicity and flexibility by enabling precise connections between any two densities over finite time intervals. Liu et al. (2023) concentrated on a linear interpolation with $\psi_t(x_0, x_1) = (1-t)x_0 + tx_1$, i.e., straight paths connecting points from the source and the target distributions. Lipman et al. (2023) introduced the interpolation through the lens of conditional probability paths leading to a Gaussian. Extensions of Lipman et al. (2023) were detailed by Tong et al. (2024), generalizing the method beyond a Gaussian source distribution. Albergo & Vanden-Eijnden (2023); Albergo et al. (2023) introduced stochastic interpolants with more general forms.

**Straightening Flows:** Liu et al. (2023) outlined an iterative process called ReFlow for coupling the points from the source and target distributions to straighten the transport path and demonstrated that repeating this procedure leads to an optimal transport map. Other related studies bypass the iterations by modifying how noise and data are sampled during training. For example, Pooladian et al. (2023); Tong et al. (2024) calculated mini-batch optimal transport couplings between the Gaussian and data distributions to minimize transport costs and gradient variance. Note that these approaches are orthogonal to our approach and can be adopted in our formulation (see Appendix H).

**Modeling Velocity Distribution:** Concurrently, Guo & Schwing (2025) also study a method to model multi-modal velocity vector fields. In this paper, we discuss use of a hierarchy of ordinary differential equations. Differently, Guo & Schwing (2025) study how to use a lower-dimensional latent space to enable modeling of the velocity distribution via a variational approach. The hierarchy of ordinary differential equations permits to more accurately model the velocity distribution while use of the variational approach enables to capture semantics.

## 6 DISCUSSION & CONCLUSION

We study a hierarchical rectified flow formulation that hierarchically couples linear ODEs, each akin to a classic rectified flow formulation. We find this formulation to accurately model multi-modal distributions for velocity, etc., which in turn enables integration paths to intersect during data generation. As a consequence, integration paths are less curved leading to compelling results with fewer neural function evaluations. Currently, our sampling process is relatively simple, relying on the Euler method for multiple integrations. We have only performed a basic grid search regarding possible integration schedules and we have not explored other solvers. We suspect, better strategies exist and we leave their exploration to future work.

**Acknowledgements:** Work supported in part by NSF grants 1934757, 2008387, 2045586, 2106825, MRI 1725729, NIFA award 2020-67021-32799, and the Alfred P. Sloan Foundation.

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

APPENDIX: TOWARDS HIERARCHICAL RECTIFIED FLOW

The appendix is organized as follows. We first provide a proof of Theorem 1 (the velocity distribution given $x_t$) in Appendix A. We then provide a proof of Corollary 1 (velocity distribution for the special case of a mixture of Gaussians target distribution) in Appendix B. Afterwards we provide the proof of Theorem 2 (correctness of the marginals) in Appendix C. Then we discuss density estimation for HRF models in Appendix D. Next we provide more details regarding the hierarchical rectified flow formulation in Appendix E. Subsequently, we discuss experimental and implementation details in Appendix F. Finally, we provide additional ablation studies in Appendix G and additional experimental results in Appendix H.

## A  PROOF OF THEOREM 1

**Proof of Theorem 1:** The velocity at location $x_t$ and time $t$ is $v = x_1 - x_0 = \frac{x_1 - x_t}{1 - t}$. The last equality holds because $(1 - t)x_0 + tx_1 = x_t$. Recall that for a random variable $Y = \alpha X + \beta$ with $\alpha, \beta \in \mathbb{R}$ and $\alpha \neq 0$, we have $p_Y(y) = \frac{1}{\alpha}p_X\left(\frac{y - \beta}{\alpha}\right)$. Since the random variable $V$ is a linear transform of the random variable $X_1$, we get

$$\pi_1(v; x_t, t) = p_{V|X_t}(v|x_t) = (1 - t)p_{X_1|X_t}((1 - t)v + x_t|x_t). \tag{13}$$

Therefore, we need to evaluate $p_{X_1|X_t}$. Using Bayes' formula,

$$p_{X_1|X_t}(x_1|x_t) = \frac{p_{X_t|X_1}(x_t|x_1)p_{X_1}(x_1)}{p_{X_t}(x_t)}, \tag{14}$$

assuming that $p_{X_t}(x_t) \neq 0$. It is undefined if $p_{X_t}(x_t) \neq 0$. Now it remains to find $p_{X_t|X_1}$ and we have

$$p_{X_t|X_1}(x_t|x_1) = p_{(1-t)X_0+tx_1}(x_t) = \frac{1}{1 - t}p_{X_0}\left(\frac{x_t - tx_1}{1 - t}\right). \tag{15}$$

Plugging Eq. (14) and Eq. (15) into Eq. (13) and using $x_1 = x_t + (1 - t)v$, we have

$$\begin{aligned}\pi_1(v; x_t, t) = p_{V|X_t}(v|x_t) &= \frac{p_{X_0}(x_t - tv)p_{X_1}(x_t + (1 - t)v)}{p_{X_t}(x_t)} \\ &= \frac{\rho_0(x_t - tv)\rho_1(x_t + (1 - t)v)}{\rho_t(x_t)}. \end{aligned} \tag{16}$$

Since the random variable $X_t$ is a linear combination of two independent random variables $X_0$ and $X_1$ as defined in Eq. (2), we have

$$\begin{aligned}\rho_t(x_t) = p_{(1-t)X_0}(x_t) * p_{tX_1}(x_t) &= \int p_{(1-t)X_0}(z)p_{tX_1}(x_t - z)dz \\ &= \int \frac{1}{1 - t}p_{X_0}\left(\frac{z}{1 - t}\right)\frac{1}{t}p_{X_1}\left(\frac{x_t - z}{t}\right)dz \\ &= \frac{1}{t(1 - t)}\rho_0\left(\frac{x_t}{1 - t}\right) * \rho_1\left(\frac{x_t}{t}\right), \quad \text{for } t \in (0, 1). \end{aligned} \tag{17}$$

At $t = 0$, $\rho_t = \rho_0$ since $x_t = x_0$. At $t = 1$, $\rho_t = \rho_1$, since $x_t = x_1$. $\pi_1(v; x_t, t)$ is undefined if $\rho_t(x_t) = 0$. This completes the proof. ∎

## B  PROOF OF COROLLARY 1

Bromiley (2003) summarizes a few useful properties for the product and convolution of Gaussian distributions. We state the relevant results here for our proof of Corollary 1.

**Lemma 1** *For the linear transform of a Gaussian random variable, we have*

$$\mathcal{N}(ax + b; \mu, \sigma^2) = \frac{1}{a}\mathcal{N}\left(x; \frac{\mu - b}{a}, \frac{\sigma^2}{a^2}\right).$$

**Lemma 2** *For the convolution of two Gaussian distributions, we have*

$$\mathcal{N}(x; \mu_1, \sigma_1^2) * \mathcal{N}(x; \mu_2, \sigma_2^2) = \mathcal{N}(x; \mu_1 + \mu_2, \sigma_1^2 + \sigma_2^2).$$

**Lemma 3** *For the product of two Gaussian distributions, we have*

$$\mathcal{N}(x; \mu_1, \sigma_1^2) \cdot \mathcal{N}(x; \mu_2, \sigma_2^2) = \frac{1}{\sqrt{2\pi(\sigma_1^2 + \sigma_2^2)}} \exp\left[-\frac{(\mu_1 - \mu_2)^2}{\sigma_1^2 + \sigma_2^2}\right] \mathcal{N}\left(x; \frac{\mu_1 \sigma_2^2 + \mu_2 \sigma_1^2}{\sigma_1^2 + \sigma_2^2}, \frac{\sigma_1^2 \sigma_2^2}{\sigma_1^2 + \sigma_2^2}\right).$$

The proofs of the Lemmas are detailed by Bromiley (2003).

**Proof of Corollary 1:** We first compute the density of $X_t$ using Theorem 1 with the specific $\rho_0$ and $\rho_1$:

$$\rho_t(x_t) = \frac{1}{t(1-t)} \rho_0\left(\frac{x_t}{1-t}\right) * \rho_1\left(\frac{x_t}{t}\right)$$

$$= \frac{1}{t(1-t)} \mathcal{N}\left(\frac{x_t}{1-t}; 0, 1\right) * \left(\sum_{k=1}^{K} w_k \mathcal{N}\left(\frac{x_t}{t}; \mu_k, \sigma_k^2\right)\right). \tag{18}$$

By applying Lemma 1 and Lemma 2 to Eq. (18), we get

$$\rho_t(x_t) = \mathcal{N}\left(x_t; 0, (1-t)^2\right) * \left(\sum_{k=1}^{K} w_k \mathcal{N}\left(x_t; t\mu_k, t^2\sigma_k^2\right)\right)$$

$$= \sum_{k=1}^{K} w_k \left(\mathcal{N}\left(x_t; 0, (1-t)^2\right) * \mathcal{N}\left(x_t; t\mu_k, t^2\sigma_k^2\right)\right)$$

$$= \sum_{k=1}^{K} w_k \mathcal{N}\left(x_t; t\mu_k, \tilde{\sigma}_{k,t}^2\right). \tag{19}$$

Using Theorem 1 and Eq. (19), we have

$$p_{V|X_t}(v|x_t) = \frac{\mathcal{N}(x_t - tv; 0, 1)\left(\sum_{k=1}^{K} w_k \mathcal{N}\left(x_t + (1-t)v; \mu_k, \sigma_k^2\right)\right)}{\sum_{k'=1}^{K} w_{k'} \mathcal{N}\left(x_t; t\mu_{k'}, \tilde{\sigma}_{k',t}^2\right)}$$

$$\stackrel{a}{=} \frac{\mathcal{N}\left(v; \frac{x_t}{t}, \frac{1}{t^2}\right)\left(\sum_{k=1}^{K} w_k \mathcal{N}\left(v; \frac{\mu_k - x_t}{1-t}, \frac{\sigma_k^2}{(1-t)^2}\right)\right)}{\sum_{k'=1}^{K} w_{k'} \mathcal{N}\left(x_t; t\mu_{k'}, \tilde{\sigma}_{k',t}^2\right)}$$

$$= \frac{\sum_{k=1}^{K} w_k \mathcal{N}\left(v; \frac{x_t}{t}, \frac{1}{t^2}\right) \mathcal{N}\left(v; \frac{\mu_k - x_t}{1-t}, \frac{\sigma_k^2}{(1-t)^2}\right)}{t(1-t) \sum_{k'=1}^{K} w_{k'} \mathcal{N}\left(x_t; t\mu_{k'}, \tilde{\sigma}_{k',t}^2\right)}$$

$$\stackrel{b}{=} \frac{\sum_{k=1}^{K} w_k \frac{t(1-t)}{\sqrt{2\pi((1-t)^2 + t^2\sigma_k^2)}} \exp\left(-\frac{(x_t - t\mu_k)^2}{(1-t)^2 + t^2\sigma_k^2}\right) \mathcal{N}\left(v; \frac{(1-t)(\mu_k - x_t) + t\sigma_k^2 x_t}{\tilde{\sigma}_{k,t}^2}, \frac{\sigma_k^2}{\tilde{\sigma}_{k,t}^2}\right)}{t(1-t) \sum_{k'=1}^{K} w_{k'} \mathcal{N}\left(x_t; t\mu_{k'}, \tilde{\sigma}_{k',t}^2\right)}$$

$$\stackrel{c}{=} \frac{\sum_{k=1}^{K} w_k \mathcal{N}\left(x_t; t\mu_k, \tilde{\sigma}_{k,t}^2\right) \mathcal{N}\left(v; \frac{(1-t)(\mu_k - x_t) + t\sigma_k^2 x_t}{\tilde{\sigma}_{k,t}^2}, \frac{\sigma_k^2}{\tilde{\sigma}_{k,t}^2}\right)}{\sum_{k'=1}^{K} w_{k'} \mathcal{N}\left(x_t; t\mu_{k'}, \tilde{\sigma}_{k',t}^2\right)}$$

$$= \sum_{k=1}^{K} \tilde{w}_{k,t} \mathcal{N}\left(v; \frac{(1-t)(\mu_k - x_t) + t\sigma_k^2 x_t}{\tilde{\sigma}_{k,t}^2}, \frac{\sigma_k^2}{\tilde{\sigma}_{k,t}^2}\right). \tag{20}$$

The equality $a$ holds by applying Lemma 1. The equality $b$ is derived by applying Lemma 3 to the product of two Gaussian distributions. Simplifying the expressions, we get equality $c$ and the final expression of $p_{V|X_t}(v|x_t)$. This completes the proof. ∎

## C    PROOF OF THEOREM 2

According to Theorem 3.3 of Liu et al. (2023), the ODE in Eq. (7) generates the samples from the ground-truth velocity distributions at space time location $(x_t, t)$. In other words, the random variable $V \sim \pi_1$.

Now we consider the characteristic function of $Z_{t+\Delta t} = Z_t + V\Delta t$ for $t \in [0, 1]$ and $\Delta t \in [0, 1-t]$, assuming that $Z_t$ has the same distribution as $X_t$. If the characteristic functions of $Z_{t+\Delta t}$ and $X_{t+\Delta t}$ agree, then $Z_{t+\Delta t}$ and $X_{t+\Delta t}$ have the same distribution.

To show this, we evaluate the characteristic function of $Z_{t+\Delta t}$,

$$
\begin{aligned}
\mathbb{E}\left[e^{\imath\langle k, Z_{t+\Delta t}\rangle}\right] &= \mathbb{E}_{\rho_t, \pi_1}\left[e^{\imath\langle k, X_t + V\Delta t\rangle}\right] \\
&= \int\int e^{\imath\langle k, x_t + v\Delta t\rangle} \pi_1(v; x_t, t)\rho_t(x_t)dvdx_t \\
&\overset{a}{=} \int\int e^{\imath\langle k, x_t + v\Delta t\rangle} \frac{\rho_0(x_t - tv)\rho_1(x_t + (1-t)v)}{\rho_t(x_t)}\rho_t(x_t)dvdx_t \\
&= \int\int e^{\imath\langle k, (x_t + v\Delta t)\rangle}\rho_0(x_t - tv)\rho_1(x_t + (1-t)v)dvdx_t \\
&\overset{b}{=} \int\int e^{\imath\langle k, (1-t-\Delta t)x_0 + (t+\Delta t)x_1\rangle}\rho_0(x_0)\rho_1(x_1)dx_0dx_1 \\
&= \mathbb{E}_{\rho_{t+\Delta t}}\left[e^{\imath\langle k, X_{t+\Delta t}\rangle}\right].
\end{aligned}
\tag{21}
$$

We use the notation $\langle \cdot, \cdot \rangle$ to denote the inner product. The equality $a$ is valid due to Theorem 1. The equality $b$ holds because $x_0 = x_t - tv$ and $x_1 = x_t + (1-t)v$ with the linear interpolation. Therefore, we find that $Z_{t+\Delta t}$ and $X_{t+\Delta t}$ follow the same distribution. In addition, since $Z_0$ and $X_0$ follow the same distribution $\rho_0$, we can conclude that $Z_t$ and $X_t$ follow the same marginal distribution at $t$ for $t \in [0, 1]$. This completes the proof.

## D    DENSITY ESTIMATION

In the following, we describe two approaches for density estimation. The resulting procedures are summarized in Algorithm 3 and Algorithm 4. To empirically verify the correctness of the density estimation procedures, we train an RF baseline and an HRF2 model using a bimodal Gaussian target distribution and a standard Gaussian source distribution (see Appendix H.1 for more details). In Fig. 7 we compare 1) the ground truth density, 2) the density estimated for the RF baseline model, and 3) the densities estimated for the HRF2 model with both procedures. We also report bits per dimension (bpd) for experiments on the 1D $1\mathcal{N} \to 2\mathcal{N}$, 2D $8\mathcal{N} \to$ moon, CIFAR-10, and ImageNet-32 data. The results are shown in Table 1. We observe that HRF2 consistently outperforms the RF baseline.

To estimate the density, according to Eq. (4) in Theorem 1, we have

$$
\log \rho_1(z_1) = \log \pi_1(u; z_t, t) + \log \rho_t(z_t) - \log \rho_0(z_t - tu), \text{ with } u = \frac{z_1 - z_t}{1-t}.
\tag{22}
$$

This implies that for any given $t \in [0, 1]$, we can use Eq. (22) to estimate the density for a generated sample $z_1$. We can choose $z_t$ using the linear interpolation in Eq. (2) with $z_0 \sim \rho_0$.

For $t = 0$, we observe that $\rho_1(z_1) = \pi_1(z_1 - z_0; z_0, 0)$, where $z_0 \sim \rho_0$. In this case, we can directly evaluate the likelihood of the generated sample via the velocity distribution. We discuss evaluation of the likelihood below. The procedure to compute the density is summarized in Algorithm 3.

For $t = 1$, the right-hand side of Eq. (22) becomes $\log \rho_1(z_1)$ because $\log \pi_1(u; z_1, 1) = \log \rho_0(z_1 - u)$, which cancels out with the last term in Eq. (22). Hence, $t = 1$ can't be used to estimate the density.

For $t \in (0, 1)$, we need to evaluate $\rho_t(z_t)$ to estimate the likelihood of $z_1$. Considering a one step linear flow from $z_0$ at time 0 to $z_t$ at $t$, we have $z_t = z_0 + vt$ and $\rho_t(z_t|z_0) = \frac{1}{t}\pi_1(v; z_0, 0)$. Using

---

**Algorithm 3:** Density Estimation 1 ($t = 0$)

---

**Input** : Generated sample $z_1$ and the source distribution $\rho_0$.

1 Sample $z_0 \sim \rho_0$ ;
2 Compute $u = z_1 - z_0$ ;
3 Compute $\hat{\rho}_1(z_1) = \pi_1(u; z_0, 0)$ according to Eq. (24) ;
4 (Optional) Compute $\hat{\rho}_1(z_1) = \frac{1}{N} \sum_{i=1}^{N} \pi_1(u^{(i)}; z_0^{(i)}, 0)$, with $u^{(i)} = z_1 - z_0^{(i)}$ and $z_0^{(i)} \sim \rho_0$ ;
   **Output:** $\hat{\rho}_1(z_1)$

---

---

**Algorithm 4:** Density Estimation 2 ($t \in (0, 1)$)

---

**Input** : Generated sample $z_1$ and the source distributions $\rho_0$ and $\pi_0$.

1 Draw random $t \sim \mathrm{Unif}(0, 1)$ ;
2 Sample $z_0 \sim \rho_0$ ;
3 Compute $z_t = t z_1 + (1 - t) z_0$ and $u = \frac{z_1 - z_t}{1 - t}$ ;
4 Evaluate $\rho_0(z_t - tu)$, $\rho_t(z_t)$ according to Eq. (23), and $\pi_1(u; z_t, t)$ according to Eq. (24) ;
5 Compute the log likelihood according to Eq. (22) ;
   **Output:** $\hat{\rho}_1(z_1)$

---

it, the density at time $t$ can be computed according to

$$\rho_t(z_t) = \int \frac{1}{t} \pi_1 \left( \frac{z_t - z_0}{t}; z_0, 0 \right) \rho_0(z_0) \, dz_0 \approx \frac{1}{N} \sum_{i=1}^{N} \frac{1}{t} \pi_1 \left( \frac{z_t - z_0^{(i)}}{t}; z_0^{(i)}, 0 \right), \qquad (23)$$

where $z_0^{(i)} \sim \rho_0$. Algorithm 4 outlines the procedure for the likelihood computation with a randomly drawn $t \in (0, 1)$. Optionally, we can average across randomly drawn $t \in (0, 1)$.

To evaluate the (log-)likelihood of a velocity $u$ at location $z_t$ and time $t$, which is needed in both cases ($t = 0$ and $t \in (0, 1)$), we follow the approach introduced by Chen et al. (2018); Song et al. (2021b) and numerically evaluate

$$\log \pi_1(u; z_t, t) = \log \pi_0(u_0; z_t, t) - \int_1^0 \nabla_{u_\tau} \cdot a_\theta(z_t, t, u_\tau, \tau) \, d\tau. \qquad (24)$$

Here, the random variable $u_\tau$ as a function of $\tau$ can be obtained by solving the ODE in Eq. (7) backward with a fixed $u$ at $\tau = 1$. The term $\nabla_{u_\tau} \cdot a_\theta(z_t, t, u_\tau, \tau)$ is computed by using the Skilling-Hutchinson trace estimator $\mathbb{E}_{p(\epsilon)} \left[ \epsilon^T \nabla_{u_\tau} a(z_t, t, u_\tau, \tau) \epsilon \right]$ (Skilling, 1989; Hutchinson, 1990; Grathwohl et al., 2018). The vector-Jacobian product $\epsilon^T \nabla_{v_\tau} a(z_t, t, u_\tau, \tau)$ can be efficiently computed by using reverse mode automatic differentiation, at approximately the same cost as evaluating $a(z_t, t, u_\tau, \tau)$.

In our experiments, we use the RK45 ODE solver (Dormand & Prince, 1980) provided by the `scipy.integrate.solve_ivp` package. We use atol $= 1e{-}5$ and rtol $= 1e{-}5$. When implementing Algorithm 4, we use $N = 1000$ to evaluate $\rho_t(x_t)$.

As mentioned above, to empirically verify the correctness of the density estimation procedures, we train an RF baseline and an HRF2 model using a bimodal Gaussian target distribution and a standard Gaussian source distribution. We compare the density estimated for the RF baseline model and the densities estimated for the HRF2 model with both Algorithm 3 and Algorithm 4. Fig. 7(a) compares the results obtained with Algorithm 3 to the RF baseline and the ground truth. Fig. 7(b) compares the density estimated for different times $t$ with Algorithm 4 to the RF baseline and the ground truth. Regardless of the choice of algorithm and time, we observe that the HRF2 model obtains a better estimation of the likelihood. Importantly, both procedures provide a compelling way to estimate densities.

In Table 1, we report bits per dimension (bpd) for experiments on the 1D $1\mathcal{N} \rightarrow 2\mathcal{N}$, 2D $8\mathcal{N} \rightarrow$ moon, CIFAR-10, and ImageNet-32 data. For 1D data, $z_0 = 0$ suffices for compelling results. For higher dimensional data, we use 20 samples of $z_0$ as shown in the optional line 4 of Algorithm 3 to compute the bits per dimension. We observe that HRF2 consistently outperforms the RF baseline.

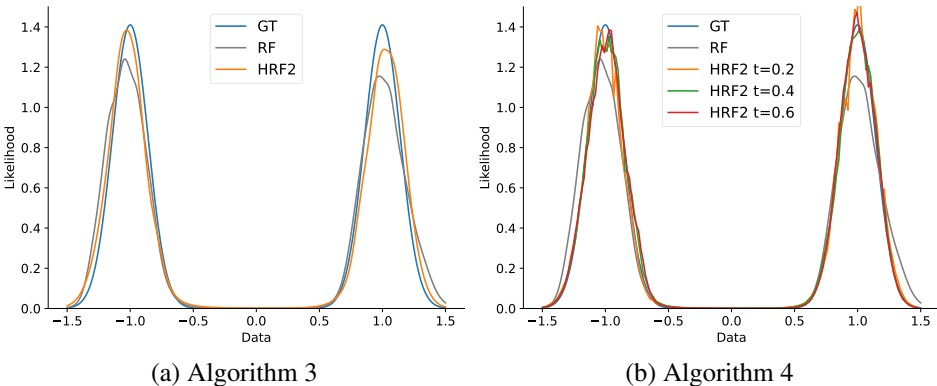

(a) Algorithm 3          (b) Algorithm 4

Figure 7: Density estimation results and comparison to ground truth. Irrespective of the choise of algorithm and the choice of time, we observe compelling density estimation results. We also note that the HRF2 model improves upon the RF baseline.

| NLL (BPD↓) | $1\mathcal{N} \to 2\mathcal{N}$ | $8\mathcal{N} \to$ moon | CIFAR-10 | ImageNet-32 |
|---|---|---|---|---|
| Baseline (RF) | 0.275 | 2.119 | 2.980 | 3.416 |
| Ours (HRF2) | 0.261 | 2.113 | 2.975 | 3.397 |

Table 1: Density estimation on 1D $1\mathcal{N} \to 2\mathcal{N}$, 2D $8\mathcal{N} \to$ moon, CIFAR-10, and ImageNet-32 data using bits per dimension (bpd). We observe a consistently better density estimation with the HRF2 model.

## E   HIERARCHICAL RECTIFIED FLOW FORMULATION DETAILS

In this section, we show how Eq. (8) can be derived from Eq. (11). For convenience we re-state Eq. (11):

$$\inf_f \mathbb{E}_{\boldsymbol{x}_0 \sim \boldsymbol{\rho}_0, x_1 \sim \rho_1, \boldsymbol{t} \sim U[0,1]^D} \left[ \left\| \left(x_1 - \mathbf{1}_D^T \boldsymbol{x}_0\right) - f\left(\boldsymbol{x_t}, \boldsymbol{t}\right) \right\|_2^2 \right]. \tag{25}$$

For $D = 2$, we note that $x_1 - \mathbf{1}_D^T \boldsymbol{x}_0$ is equivalent to $x_1 - x_0^{(1)} - x_0^{(2)}$. Letting $x_0 = x_0^{(1)}$ and $v_0 = x_0^{(2)}$, we obtain $x_1 - \mathbf{1}_D^T \boldsymbol{x}_0 = x_1 - x_0 - v_0$.

Further note that we obtain the time variables $\boldsymbol{t} = [t^{(1)}, t^{(2)}] = [t, \tau] \sim U[0,1]^2$, since $t$ and $\tau$ are drawn independently from a uniform distribution $U[0,1]$. Also, $\boldsymbol{x}_0 = [x_0^{(1)}, x_0^{(2)}] = [x_0, v_0] \sim \boldsymbol{\rho}_0$, where $x_0$ and $v_0$ are drawn independently from standard Gaussian source distributions $\rho_0$ and $\pi_0$ because $\boldsymbol{\rho}_0$ is a $D$-dimensional standard Gaussian.

Based on the general expression $x_{\boldsymbol{t}}^{(d)} = (1 - t^{(d)})x_0^{(d)} + t^{(d)}(x_1 - \sum_{k=1}^{d-1} x_0^{(k)})$ and the previous results, we have $x_t = x_{\boldsymbol{t}}^{(1)} = (1 - t^{(1)})x_0^{(1)} + t^{(1)}x_1 = (1 - t)x_0 + tx_1$ and $v_\tau = x_{\boldsymbol{t}}^{(2)} = (1 - t^{(2)})x_0^{(2)} + t^{(2)}(x_1 - x_0^{(1)}) = (1 - \tau)v_0 + \tau v_1$. This is identical to the computation of $x_t$ and $v_\tau$. Combining all of these results while renaming the function from $f$ to $a$, we arrive at

$$\inf_a \mathbb{E}_{x_0 \sim \rho_0, x_1 \sim \mathcal{D}, t \sim U[0,1], v_0 \sim \pi_0, \tau \sim U[0,1]} \left[ \left\| (x_1 - x_0 - v_0) - a(x_t, t, v_\tau, \tau) \right\|_2^2 \right]. \tag{26}$$

This program is identical to the one stated in Eq. (8).

## F   EXPERIMENTAL AND IMPLEMENTATION DETAILS

### F.1   LOW DIMENSIONAL EXPERIMENTS

For the 1D and 2D experiments, we use the same neural network. It consists of two parts. The first part processes the space and time information separately using sinusoidal positional embedding and

linear layers. In the second part, the processed information is concatenated and passed through a series of linear layers to produce the final output. Compared to the baseline, our HRF model with depth $D$ takes $D$ times more space and time information as input. Therefore, the first part of the network has $D$ times more embedding and linear layers to handle spatial and temporal information from different depths. However, by adjusting the dimensions of the hidden layers, we reduced the network size to just one-fourth of the baseline, while achieving superior performance. For each dataset in the low-dimensional experiments, we use 100,000 data points for training and another 100,000 data points for evaluation. For each set of experiments, we train five different models using five random seeds. During the evaluation, we performed a total of 125 experiments and averaged the results to ensure the fairness and validity of our findings.

## F.2 HIGH DIMENSIONAL EXPERIMENTS

In the high-dimensional image experiments, we used the U-Net architecture described by Lipman et al. (2023) for the baseline model. To handle extra inputs $v_\tau$ and $\tau$, we designed new U-Net-based network architectures for MNIST, CIFAR-10, and ImageNet-32 data.

**MNIST.** For MNIST, we use a single U-Net and modify the ResNet block. Similar to the neural network used in our low-dimensional experiments, each ResNet block has two parts. In the first part, we handle two sets of space-time information, i.e., $(x_t, t)$ and $(v_\tau, \tau)$, separately with 2 distinct pathways: convolutional layers for spatial data and linear layers for time embeddings. In the second part, all the spatial data and time embeddings are added together and passed through a series of linear layers to capture the space-time dependencies. For a fair evaluation, we adjusted the number of channels such that the model sizes approximately match (ours: 1.07M parameters vs. baseline: 1.08M parameters). We note that the HRF formulation significantly outperforms the baseline. The results were shown in Fig. 6. More results are provided in Appendix H.3.

**CIFAR-10.** For CIFAR-10, we use two U-Nets with the same number of layers but different channel sizes. We use a larger U-Net with channel size 128 to process the velocity $v_\tau$ and time $\tau$. We use another smaller U-Net with channel size 32 to process the location $x_t$ and time $t$. We merge the output of each ResNet block of the smaller U-Net with the corresponding ResNet block of the bigger U-Net. The size of this new U-Net structure is $1.25\times$ larger than the baseline (44.81M parameters in our model and 35.75M parameters in the baseline). Our model achieves a slightly better generation quality (see Fig. 6 in Section 4 and Table 7 in Appendix H.3).

**ImageNet-32.** For ImageNet-32, we adopt the same architectural setup as for CIFAR-10 but modify the attention resolution to "16,8" instead of just "16" to better capture the increased multimodality of the ImageNet-32 dataset. Our U-Net model has a parameter size of 46.21M, compared to 37.06M for the baseline. It demonstrates slightly improved generation quality (see Fig. 6 in Section 4 and Table 7 in Appendix H.3).

For training, we adopt the procedure and parameter settings from Tong et al. (2024) and Lipman et al. (2023). We use the Adam optimizer with $\beta_1 = 0.9$, $\beta_2 = 0.999$, and $\epsilon = 10^{-8}$, with no weight decay. For MNIST, the U-Net has channel multipliers $[1, 2, 2]$, while for CIFAR-10 and ImageNet-32, the channel multipliers are $[1, 2, 2, 2]$. The learning rate is set to $1 \times 10^{-4}$ with a batch size 128 for MNIST and CIFAR-10. For ImageNet-32, we increase the batch size to 512 and adjust the learning rate to $2 \times 10^{-4}$. We train all models on a single NVIDIA RTX A6000 GPU. For MNIST, we train both the baseline and our model for 150,000 steps while we use 400,000 steps for CIFAR-10.

# G ABLATION STUDIES

## G.1 ABLATION STUDY FOR NFE

The sampling process of HRF with depth $D$ involves integrating $D$ ODEs using Euler's method. The total number of neural function evaluations (NFE) is defined as NFE $= \prod_d N^{(d)}$ where $N^{(d)}$ is the number of integration steps at depth $d$. Note, for a constant NFE budget, varying the $N^{(d)}$ values can lead to different results. Therefore, we conduct an ablation study to understand suitable choices for $N^{(d)}$.

| Total NFEs | Sampling Steps | $\mathcal{N} \to 2\mathcal{N}$ 1-WD | $\mathcal{N} \to 5\mathcal{N}$ 1-WD | $2\mathcal{N} \to 2\mathcal{N}$ 1-WD | $\mathcal{N} \to 6\mathcal{N}(2D)$ 2-SWD | $8\mathcal{N} \to \text{moon}$ 2-SWD |
|---|---|---|---|---|---|---|
| 100 | $(1, 100)$ | **0.020** | 0.031 | 0.045 | 0.070 | 0.172 |
| 100 | $(2, 50)$ | 0.025 | 0.019 | 0.011 | **0.037** | **0.107** |
| 100 | $(5, 20)$ | 0.022 | 0.020 | **0.010** | 0.045 | 0.119 |
| 100 | $(10, 10)$ | 0.025 | 0.019 | 0.017 | 0.053 | 0.163 |
| 100 | $(20, 5)$ | 0.026 | **0.017** | 0.030 | 0.062 | 0.201 |
| 100 | $(50, 2)$ | 0.047 | 0.030 | 0.075 | 0.081 | 0.222 |
| 100 | $(100, 1)$ | 0.032 | 0.030 | 0.050 | 0.085 | 0.177 |

Table 2: HRF2 performance for low dimensional experiments under the same NFE $= 100$ budget with different choices of sampling steps. Sampling steps $(J, L)$ indicates that we use $J$ steps to integrate $x$ and $L$ steps to integrate $v$. 1-WD refers to the 1-Wasserstein distance and 2-SWD refers to the Sliced 2-Wasserstein distance. **Bold** for the best. Underline for the runner-up.

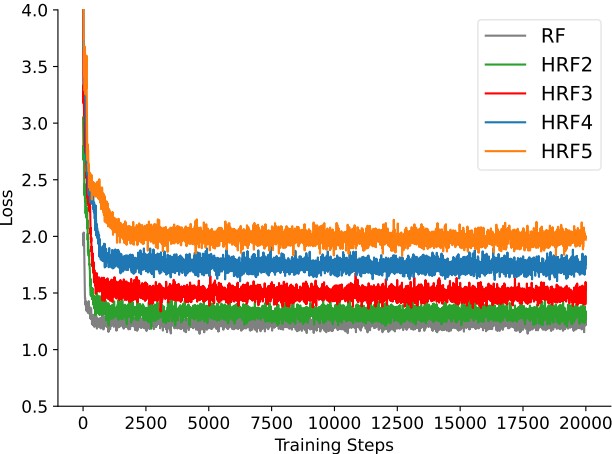

Figure 8: Training losses of HRF with different depths on 1D data, a standard Gaussian source distribution to a mixture of 2 Gaussians target distribution. We observe training to remain stable.

As shown in Fig. 3, increasing the number of integration steps improves the sampling of the velocity distribution. However, beyond a certain threshold, the benefit of additional steps does not justify the increased computational cost. Table 2 further illustrates that, for a fixed NFE budget, a compelling strategy is to allocate a sufficient number of steps to accurately sample $v$ for a precise velocity distribution while using fewer steps to integrate over $x$.

### G.2 ABLATION STUDY FOR DEPTH

Our HRF framework can be extended to an arbitrary depth $D$. Here, we compare the training loss convergence of HRF with depths ranging from 1 to 5, where HRF1 corresponds to the baseline RF. As illustrated by the training losses shown in Fig. 8, training stability remains consistent across different depths, with higher-depth HRFs demonstrating comparable stability to lower-depth models. Importantly, note that Fig. 8 mainly serves to compare convergence behavior and not loss magnitudes as those magnitudes reflect different objects, i.e., velocity for a depth of 1, acceleration for a depth of 2, etc. Moreover, the deep net structure for the functional field of directions $f$ depends on the depth, which makes a comparison more challenging. Table 3 and Table 4 indicate that increasing the depth results in manageable model size, training time, and inference time. These trade-offs are justified by the significant performance improvements observed in Fig. 4 and Fig. 5. See Appendix H.1 for details regarding the training data.

| Training | 1D data | | | 2D data | | |
|---|---|---|---|---|---|---|
| | RF (0.30M) | HRF2 (0.07M) | HRF3 (0.67M) | RF (0.33M) | HRF2 (0.08M) | HRF3 (0.71M) |
| Time ($\times 10^{-2}$ s/iter) | 1.292 | 0.736 | 2.202 | 1.503 | 0.737 | 2.252 |
| Memory (MB) | 2011 | 1763 | 2417 | 2091 | 1803 | 2605 |
| Param. Counts | 297,089 | 74,497 | 673,793 | 329,986 | 76,674 | 711,042 |

Table 3: Computational requirements for training on synthetic datasets. All models in this table are trained for 15000 iterations with a batch size of 51200.

| Inference Time (s) | 1D data | | | 2D data | | |
|---|---|---|---|---|---|---|
| Total NFEs | RF (0.30M) | HRF2 (0.07M) | HRF3 (0.67M) | RF (0.33M) | HRF2 (0.08M) | HRF3 (0.71M) |
| 5 | 0.030 ± 0.014 | 0.014 ± 0.005 | 0.037 ± 0.030 | 0.035 ± 0.017 | 0.017 ± 0.006 | 0.041 ± 0.034 |
| 10 | 0.069 ± 0.020 | 0.033 ± 0.000 | 0.128 ± 0.001 | 0.078 ± 0.025 | 0.039 ± 0.000 | 0.145 ± 0.001 |
| 50 | 0.372 ± 0.024 | 0.164 ± 0.000 | 0.642 ± 0.001 | 0.440 ± 0.001 | 0.193 ± 0.000 | 0.727 ± 0.001 |
| 100 | 0.755 ± 0.001 | 0.327 ± 0.000 | 1.291 ± 0.002 | 0.884 ± 0.001 | 0.385 ± 0.000 | 1.455 ± 0.003 |

Table 4: Inference time comparison for synthetic data using a varying NFE budget. For HRF2, we used sampling step combinations: $(1,5), (2,5), (5,10), (10,10)$. For HRF3, we used sampling step combinations: $(1,1,5), (1,2,5), (1,5,10), (2,5,10)$. For all experiments, we set our batch size to 100,000.

# H  ADDITIONAL EXPERIMENTAL RESULTS

## H.1  ADDITIONAL 1D RESULTS

The results for experiments used in Fig. 1 and Fig. 3 are shown in Fig. 9.

## H.2  HIERARCHICAL RECTIFIED FLOW WITH OTCFM

As mentioned in Section 5, various approaches for straightening the paths in flow matching models exist. These approaches are orthogonal to our work and can be easily incorporated in the HRF formulation. To demonstrate this, we incorporate the minibatch optimal transport conditional flow matching (OTCFM) (Tong et al., 2024) into the two layered hierarchical rectified flow (HRF2). In OTCFM, for each batch of data $(\{x_0^{(i)}\}_{i=1}^B, \{x_1^{(i)}\}_{i=1}^B)$ seen during training, we sample pairs of points from the joint distribution $\gamma_{\text{batch}}(x_0, x_1)$ given by the optimal transport plan between the source and target points in the batch. We follow the same procedure to couple noise with the data points and use the batch-wise coupled $x_0$ and $x_1$ to learn the parameters in $a_\theta$. We refer to this approach as HOTCFM2. We test its performance on two synthetic examples: 1) a 1D example with a standard Gaussian source distribution and a mixture of two Gaussians as the target distribution; and 2) a 2D example with a mixture of eight Gaussians as the source distribution and the moons dataset as the target distribution. Fig. 10 and Fig. 11 show that hierarchical rectified flow improves the performance of OTCFM.

## H.3  ADDITIONAL RESULTS ON MNIST, CIFAR-10, AND IMAGENET-32

Here we show additional results for experiments with MNIST, CIFAR-10, and ImageNet-32 data. From Tables 5 to 7, we can observe the following: For MNIST, our model is comparable in size, comparable in training times, and comparable in inference times, while outperforming the baseline. For CIFAR-10 and ImageNet-32, our model is $1.25\times$ larger and has a slower inference time. However, as shown in Table 7, it still outperforms the baseline. We believe that the modest trade-off in model size and inference time is acceptable given the performance gains.

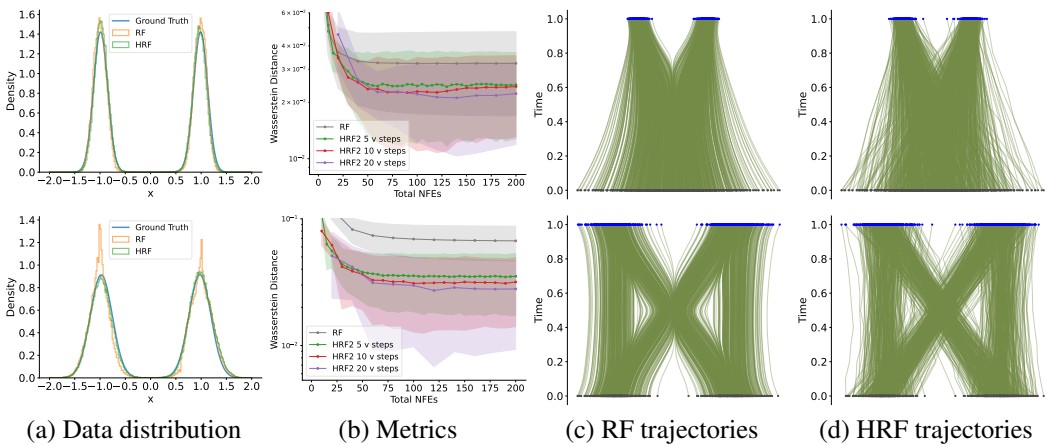

(a) Data distribution     (b) Metrics     (c) RF trajectories     (d) HRF trajectories

Figure 9: More experiments on 1D data: top row shows results for a standard Gaussian source distribution and a mixture of 2 Gaussians target distribution; bottom row shows results for a mixture of 2 Gaussians source distribution and the same mixture of 2 Gaussians target distribution.

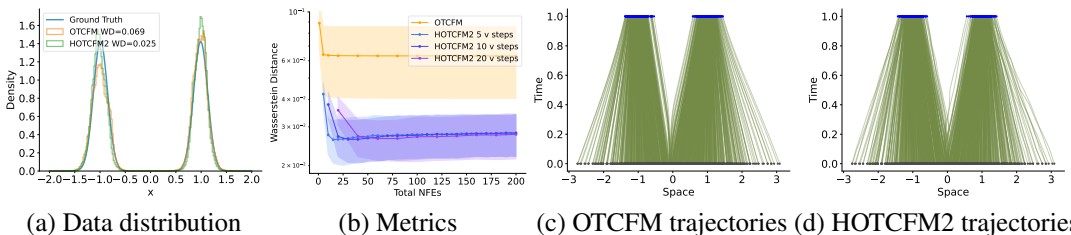

(a) Data distribution    (b) Metrics    (c) OTCFM trajectories (d) HOTCFM2 trajectories

Figure 10: Results for 1D data, with $\rho_0$ being a standard Gaussian and $\rho_1$ being a mixture of 2 Gaussians. (a) Histograms of generated samples and $\rho_1$. (b) The 1-Wasserstein distance vs. total NFEs. (c,d) The trajectories of particles flowing from source distribution (grey) to target distribution (blue).

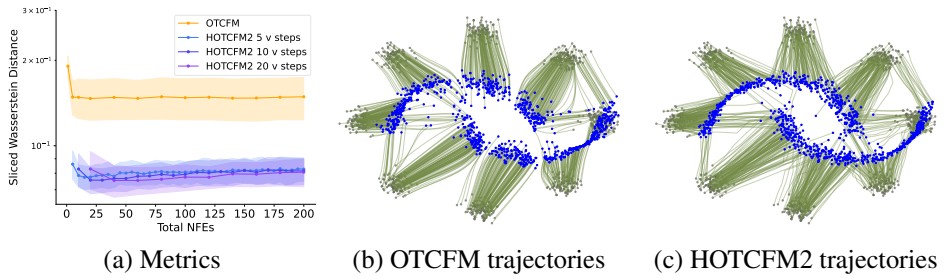

(a) Metrics     (b) OTCFM trajectories     (c) HOTCFM2 trajectories

Figure 11: Results for 2D data, with $\rho_0$ being a mixture of 8 Gaussians and $\rho_1$ being represented by the moons data. (a) Sliced 2-Wasserstein distance vs. total NFEs. (b) and (c) show the trajectories (green) of sample particles flowing from source distribution (grey) to target distribution (blue).

| Training | MNIST | | CIFAR-10 | | ImageNet-32 | |
|---|---|---|---|---|---|---|
| | RF (1.08M) | HRF2 (1.07M) | RF (35.75M) | HRF2 (44.81M) | RF (37.06M) | HRF2 (46.21M) |
| Time (s/iter) | 0.1 | 0.1 | 0.3 | 0.4 | 0.7 | 0.8 |
| Memory (MB) | 3935 | 3931 | 8743 | 10639 | 27234 | 33838 |
| Param. Counts | 1,075,361 | 1,065,698 | 35,746,307 | 44,807,843 | 37,064,707 | 46,210,083 |

Table 5: Computational requirements during training on image datasets.

| Inference time (s) | MNIST | | CIFAR-10 | | ImageNet-32 | |
|---|---|---|---|---|---|---|
| Total NFEs | RF (1.08M) | HRF2 (1.07M) | RF (35.75M) | HRF2 (44.81M) | RF (37.06M) | HRF2 (46.21M) |
| 5 | 0.084 ± 0.001 | 0.085 ± 0.001 | 0.221 ± 0.000 | 0.295 ± 0.000 | 0.229 ± 0.000 | 0.301 ± 0.000 |
| 10 | 0.168 ± 0.000 | 0.169 ± 0.000 | 0.441 ± 0.001 | 0.589 ± 0.001 | 0.458 ± 0.000 | 0.601 ± 0.000 |
| 20 | 0.336 ± 0.000 | 0.339 ± 0.000 | 0.889 ± 0.001 | 1.176 ± 0.001 | 0.918 ± 0.001 | 1.207 ± 0.001 |
| 50 | 0.843 ± 0.001 | 0.851 ± 0.002 | 2.229 ± 0.001 | 2.953 ± 0.004 | 2.302 ± 0.002 | 3.029 ± 0.004 |
| 100 | 1.693 ± 0.002 | 1.706 ± 0.003 | 4.471 ± 0.004 | 5.921 ± 0.003 | 4.618 ± 0.003 | 6.100 ± 0.014 |
| 500 | 8.538 ± 0.030 | 8.598 ± 0.010 | 22.375 ± 0.011 | 29.701 ± 0.011 | 23.110 ± 0.005 | 30.863 ± 0.083 |

Table 6: Inference time comparison for MNIST, CIFAR-10, and ImageNet-32 datasets using a varying total NFEs budget. For HRF2 on MNIST we used sampling step combinations: $(1,5), (2,5), (5,4), (5,10), (5,20), (5,100)$. For HRF2 on CIFAR-10 and ImageNet-32 we used sampling step combinations: $(1,5), (1,10), (1,20), (1,50), (2,50), (2,250)$. All experiments are conducted with a batch size of 128.

| Performance (FID) | MNIST | | CIFAR-10 | | ImageNet-32 | |
|---|---|---|---|---|---|---|
| Total NFEs | RF (1.08M) | HRF2 (1.07M) | RF (35.75M) | HRF2 (44.81M) | RF (37.06M) | HRF2 (46.21M) |
| 5 | 19.187 ± 0.188 | **15.798 ± 0.151** | 36.209 ± 0.142 | **30.884 ± 0.104** | 69.233 ± 0.166 | **48.933 ± 0.177** |
| 10 | 7.974 ± 0.119 | **6.644 ± 0.076** | 14.113 ± 0.092 | **12.065 ± 0.024** | 21.744 ± 0.045 | **20.286 ± 0.022** |
| 20 | 6.151 ± 0.090 | **3.408 ± 0.076** | 8.355 ± 0.065 | **7.129 ± 0.027** | **12.411 ± 0.002** | 12.492 ± 0.100 |
| 50 | 5.605 ± 0.057 | **2.664 ± 0.058** | 5.514 ± 0.034 | **4.847 ± 0.028** | **8.910 ± 0.137** | 9.024 ± 0.112 |
| 100 | 5.563 ± 0.049 | **2.588 ± 0.075** | 4.588 ± 0.013 | **4.334 ± 0.054** | 7.873 ± 0.110 | **7.679 ± 0.022** |
| 500 | 5.453 ± 0.047 | **2.574 ± 0.121** | 3.887 ± 0.035 | **3.706 ± 0.043** | 6.962 ± 0.087 | **6.503 ± 0.035** |

Table 7: Performance comparison for MNIST, CIFAR-10, and ImageNet-32 datasets using a varying total NFEs budget. For HRF2 on MNIST we used sampling step combinations: $(5,1), (10,1), (5,4), (10,5), (10,10), (100,5)$. For HRF2 on CIFAR-10 and ImageNet-32 we used sampling step combinations: $(1,5), (1,10), (1,20), (1,50), (2,50), (2,250)$. **Bold** for lower mean.

