# OpenReview forum: "Towards Hierarchical Rectified Flow"
_ICLR.cc/2025/Conference — ICLR 2025 Poster_

### Official Review · Reviewer_4Etx · 2024-10-28

**Soundness:** 3
**Presentation:** 3
**Contribution:** 2
**Rating:** 8
**Confidence:** 4

**Summary:**

Flows $p_t$ transform one distribution $p_0$ into another $p_1$ via a corresponding vector field $f(x_t,t)$ that satisfies its continuity equation. The flow is typically defined via conditional flows, for example $p_t(x|x_0, x_1)\sim t(x_1)+(1-t)x_0+\sigma\epsilon$, where one can take the limit $\sigma\rightarrow 0$. The corresponding flow is simply $p_t(x)=\int p_t(x|x_0, x_1) \pi(x_0, x_1)dx_0 dx_1$. The conditional vector fields $v_t(x_t|x_0,x_1)$ that satisfy the continuity equation of such flows are $x_1-x_0$, while the unconditional vector field is $f(x_t,t)=\int v_t(x_t|x_0,x_1) p_t(x_0, x_1|x_t) dx_0 dx_1=\int v_t(x_t) \pi_t(v_t|x_t) dv_t$.

This paper studies the possibility of using a sample of $\pi_t(v_t|x_t)$ at each integration step during data generation, instead of using the mean $f(x_t,t)$ which is akin to using stochastic gradient descent instead of gradient descent. . The result is a stochastic sampling process, which by using a derivation of $\pi_t(v_t|x_t)$ is proven to have marginals that coincide with the flow $p_t$.

The distribution $\pi_t(v_t|x_t)$ is itself modeled using the methodology of rectified flow matching, and the methodology is applied to synthetic and high dimensional image data.

**Strengths:**

The paper is generally well written, even though a few parts could be improved.

Theorem 1 and 2 are interesting results, and provide insights about the conditional distribution of velocities at each current point $x_t$ and they enable using higher order stochastic sampling.

**Weaknesses:**

My **main** concerns are the following:

1. The proposed method requires taking the position and the current velocity as input in order to predict the acceleration. The authors mention expanding the Resnet for their framework and increasing the amount of data processed. Can authors provide a detailed table comparing HRF and RF models, including parameter counts, training time per iteration, and memory usage across all experiments?

2. It is not clear if the NFE includes the steps required to sample the velocities. The paper should report both L and J in each experiment. Also the compute time for each step should be reported and compared with RF.

3. The results for CIFAR-10 unfortunately are not encouraging. Considering the results on MNIST and CIFAR-10 it seems that the proposed model does not scale as well. This is why it is important to also test the model on Imagenet.

4. The resulting model is not a diffeomorphism and as such we lose the ability to perform density estimation.

In my opinion the main contributions of the work are theoretical, and applying the proposed methodology in practice is challenging.

**Questions:**

1. What are the parameter counts for both HRF and the baseline HF? Also what is the training time per iteration?

2. Is the reported $NFE=J\cdot L$? What is the time required per NFE in both HFR and RF?

3. The paper claims that the resulting paths are straight as the trajectories can intersect. However the trajectories are stochastic, and thus they are likely to increase overall length. I would appreciate a clarification from the authors regarding this matter.

---

> ### Author Response · Authors · 2024-11-20
> **Response to Reviewer 4Etx**
>
> Thanks a lot for your time and feedback and for assessing the paper as presenting interesting theorems.
>
> ***QC1: Can authors provide a detailed table comparing HRF and RF models, including parameter counts, training time per iteration, and memory usage across all experiments?***
> > For synthetic data, we included details regarding the training/inference time in the newly added Appendix G (see Table 2 and Table 3). For MNIST and CIFAR-10, we compared training/inference time and generation performance of the baseline RF and of our method in the newly added Appendix H.3 of the revised appendix (see Tables 4-6). We briefly summarize the results here. For MNIST, our model outperforms the baseline while maintaining a comparable model size, training time, and inference time. For CIFAR-10, although our model is 1.25x larger and has a roughly 1.4x slower inference time, it still achieves superior performance compared to the baseline: 3.713 FID vs. 3.927 FID for the baseline. We think this trade-off is justified.
>
> ***QC2: Is the reported NFE=J⋅L? What is the time required per NFE in both HRF and RF?***
> > Yes, for HRF with depth 2, NFE=J⋅L. Note, all NFEs in our paper are Total NFEs: $\text{NFE}=\prod_d N^{(d)}$, where $N^{(d)}$ refers to the integration steps at depth $d$. To clarify, we added L349 in the revised paper. Further, all figure axis labels have been updated to “Total NFEs”. To clarify further, we added an ablation study in Appendix G, analyzing NFEs and the impact of integration steps across depths. For synthetic data, training and inference times are reported in newly added Table 2 and newly added Table 3. For MNIST and CIFAR-10, training and inference times are reported in newly added Table 4 and newly added Table 5. Summarizing the tables briefly, training and inference times are on par.
>
> ***QC3: The results for CIFAR-10 unfortunately are not encouraging. Considering the results on MNIST and CIFAR-10 it seems that the proposed model does not scale as well.***
> > We kindly disagree, on both MNIST and CIFAR-10, our approach achieves better performance using the same total NFEs. For MNIST, our model size is comparable (slightly smaller) while the results are better (see Fig. 6). For CIFAR-10, our model size is 1.25x larger but we achieve a slightly better FID of 3.713 vs. 3.927 (see Table 6 in the revised appendix). We include more details in Appendix H.3 (see Tables 4-6) and updated Figure 6 in the main paper to provide the parameter count.
>
> ***QC4: ImageNet results***
> > Unfortunately we don’t have the computational resources to conduct experiments on higher-dimensional data like ImageNet. However, our core idea---the hierarchical structure---is theoretically applicable to any (latent) diffusion architecture that utilizes flow matching for learning a vector field. We are open to exploring such experiments in future work and are very excited to collaborate with any interested parties.
>
>
> ***QC5: The resulting model is not a diffeomorphism and as such we lose the ability to perform density estimation.***
> > We added Appendix D in the revised version of the paper to show how density estimation can be performed for an HRF model. We observe our model to lead to more accurate density estimation results than the RF baseline (see newly added Figure 7).
>
>
> ***QC6: The paper claims that the resulting paths are straight as the trajectories can intersect. However the trajectories are stochastic, and thus they are likely to increase overall length. I would appreciate a clarification from the authors regarding this matter.***
> > In the paragraph starting at L291, we mentioned that our generation process for the data can be piecewise straight and that a large numerical integration step size $\Delta t$ is acceptable. For inference, piecewise straightness matters more than the overall length, because this implies that we can use less numerical integration steps while maintaining accuracy. This reduces the number of NFEs and improves the efficiency. In our experiments, we typically only use 2-5 data integration steps. Results reported in the newly added Tables 3, 5, 6 corroborate this.

---

> ### Comment · Reviewer_4Etx · 2024-11-21
>
> I thank the authors for the response. My concerns regarding computational demands have been considerably reduced. However some issues still linger.
>
> **QC1:**
>
> Time results in Table 2 are not that informative. The authors should use scientific notation to present results as the ratio between time is what matters. Or they can present the overall training time per number of iterations (fractional form), as that is the quantity I requested.
>
> I am not sure why the the same network size was not used in CIFAR experiments like in the MNIST case. That would have enabled a more direct comparison.
>
> **QC2:**
>
> This was a key concern. The fact that the total number of of steps in both methods is the same indicates that the method is not as computationally demanding as I originally feared.
>
> **QC3 & 4:**
>
> Unfortunately, we indeed kindly disagree. On MNIST the proposed method significantly outperforms the baseline using the same network size. For CIFAR a larger network is used but the difference with the baseline shrinks. As such it is important to show that the method scales. The type of scaling I am worried most relates to the increase of distribution complexity and not dimensionality. As the distribution of the data becomes more complex (has more modes and greater variability), the distribution of the velocities, as authors show, also becomes more complex. As such, at each integration step, there could be contradictory directions as a result of sampling from a highly varying distribution. Therefore it is important to test on Imagenet, even at a reduced scale of 32x32. This relates to QC6.
>
> **QC5:**
>
> I mostly meant a path-wise density estimation method. That is, a method that is computationally cheap and that does not rely on Monte-Carlo estimations. As such I prefer Algorithm 3 considerably more than 4. as it does not rely on Equation 22. Regarding Algorithm 3, is there a reason to favor also $\pi_1(0; z_1, 0)$ instead of  $\pi_1(z_1-z_0; z_0, 0)$  in Step 3? Then one can calculate
>
> $\log \pi_1(0; x, 0) = \log \pi_0(u_0; x, 0) - \int_1^0 \nabla_{u_\tau} \cdot a_\theta(x, 0, u_\tau, \tau) d\tau$
>
> using the instantaneous change of variable formula of [1] (this reference should be added). So my question is, what role does the sampling of $z_0$ play in the variability? Also reference [2] should be added when using the Hutchinson trace estimator.
>
> From an experimental perspective, I was hoping to see this method applied to 2D complex toy datasets and, more importantly, to observe bits-per-dimension results on image datasets such as MNIST and CIFAR-10. Conducting such experiments would not be particularly computationally demanding and could likely be completed within a few hours on a consumer-grade GPU.
>
> **QC6:**
> Since all generated paths are discrete approximations, they are inherently piecewise. As I clarified in QC3 & 4, my concern is that for more complex distributions, the model may struggle to decide on a direction, leading to a pronounced zig-zag pattern resembling Brownian motion, significantly more than in the baseline case. For instance, in Figure 1c, some particles appear to move from the bottom-right toward the upper-left, only to reverse course and head toward the upper-right. My concern is that this behavior could become more pronounced as the dimensionality increases and, more critically, as the distribution's complexity grows, potentially amplifying directional uncertainty.
>
> **Notation:**
> I noticed that Theorem 2 is proven for 1D data, as such $k$ should be a vector, and $kx$ should be written as $k^Tx$.
>
> Overall, I believe the appropriate evaluation for this paper is a score of 7. If the main concerns regarding scaling on ImageNet 32x32 and density estimation for at least MNIST are addressed effectively, I would consider the paper deserving of an 8. However, since selecting a score of 7 is not an option, I will assign an 8, recognizing that the paper introduces a valuable new branch for research.
>
> **References**
>
> Neural ODEs, Chen et al 2018
>
> FFJORD: Free-form Continuous Dynamics for Scalable Reversible Generative Models. Grathwohl et al. 2018.

---

> > ### Author Response · Authors · 2024-11-26
> > **Response to Reviewer 4Etx**
> >
> > Thanks a lot for your timely reply and your valuable suggestions.
> >
> > ***QC1: Table 2 training time & CIFAR-10 network structure different from MNIST’s.***
> >
> > > We adjusted the training time presented in Table 3 (previous Table 2). For the CIFAR-10 dataset, we initially experimented with the model structure used for MNIST. However, it resulted in a large model size. For a more fair comparison, we designed a new structure tailored to the increased complexity of the CIFAR-10 dataset.
> >
> >
> > ***QC2: The fact that the total number of steps in both methods is the same indicates that the method is not as computationally demanding as I originally feared.***
> >
> > > We apologize for the initial confusion and are glad to see that this could be resolved.
> >
> >
> > ***QC3 & 4: As the distribution of the data becomes more complex (has more modes and greater variability), the distribution of the velocities, as authors show, also becomes more complex. As such, at each integration step, there could be contradictory directions as a result of sampling from a highly varying distribution. Results with downsampled ImageNet.***
> >
> > > Note, Corollary 1 shows that as time $t$ approaches 1, the velocity distribution becomes more and more uni-modal. This is also illustrated in Figure 2. We think it is compelling to be able to sample from various directions when time $t$ is small. Further, this enables us to model crossing paths.
> >
> > > We recognize the interest in ImageNet results. We are running some experiments. Experimentation is slow but we’ll provide an update on the status prior to the author response deadline.
> >
> > ***QC5: Density estimation. What role does the sampling of z0 play in the variability? Additional reference.***
> >
> > > The newly added Table 1 shows the bits per dimension for different datasets. We observe HRF2 to consistently achieve competitive results. We tried your recommended $\pi_1(0;z_1,0)$ and found the performance to be lower than the reported results. For 1D data, $z_0=0$ suffices for compelling results. For higher dimensional data we use $N=20$ $z_0$ as shown in the optional line 4 of Algorithm 3 to compute the bits per dimension. We provide those details in the revised Appendix D.
> >
> > > Thanks a lot for suggesting to add references which are included in the revised version.
> >
> > ***QC6: the model may struggle to decide on a direction, leading to a pronounced zig-zag pattern resembling Brownian motion, significantly more than in the baseline case***
> >
> > > The process of our data generation follows a random differential equation with a random velocity field. A zig-zag pattern is hence natural. Note, this differs from the SDE formulation commonly used in diffusion models, where the sample path is governed by a deterministic drift term.
> >
> > > However, Corollary 1 shows that as time $t$ approaches 1, the velocity distribution becomes more and more uni-modal. This is also illustrated in Figure 2: as time  $t$ changes from 0 to 1, we observe the probability of drawing a velocity that points to another mode to decrease and the velocity distribution to become more unimodal. Therefore, we feel that HRF does not critically amplify the directional uncertainty.
> >
> > ***QC7: Theorem 2 proof for vectors***
> > > Thanks for pointing this out. We changed the notation in the revised Appendix C.

---

> ### Comment · Reviewer_4Etx · 2024-11-27
>
> Thank you for the response!
>
> **QC5:**
>
> Thank you for the additional experiments. I believe attention should be paid to MNIST results. Bits per dim scores are above 2, which is significantly higher than that of FFJORD 2018 (bits per dim of 0.99). CIFAR results are quite typical however. Also, as an advice for the final version of the paper, in the case of 2D data, authors could add 2D density plots like in Figure 2 of FFJORD 2018.
>
> **QC6:**
>
> Considering that at time $t=0$ the distribution of velocities coincides with a shifted version of the data distribution, there is a possibility that sampling stochasticity will be high when the data distribution is complex. However, this remains to be tested more thoroughly in the future.
>
> Thanks gain for the response, I look forward to seeing the Imagenet results.

---

> > ### Author Response · Authors · 2024-12-02
> > **Response to Reviewer 4Etx**
> >
> > Thanks a lot for your very thoughtful, insightful, and helpful feedback, and thanks also for your support.
> >
> > ***QC5: MNIST & plotting***
> > > Thanks a lot for the plotting suggestion, which we’ll incorporate. Thanks also for pointing out MNIST. We are aware of the differences and assessing the reasons.
> >
> > ***QC6: Sampling stochasticity***
> > > We think an adequate sampling stochasticity at early integration steps even for complex data is beneficial. We concur that there are exciting opportunities for future research, which are beyond the scope of the current paper.
> >
> > ***QC7: Current ImageNet results***
> > > We are still running ImageNet 32x32 experiments but wanted to keep the reviewer in the loop and share our current progress while the reviewer is still able to respond via openreview. Current FID results with current sampling schedules given in parenthesis are as follows:
> >
> >
> > | NFE | 5 | 10 | 20 | 50 | 100 | 500 |
> > | --- | --- | --- | --- | --- | --- | --- |
> > | Baseline | 69.5 | 22.2 | 12.7 | 9.41 | 8.50 | 7.55 |
> > | Ours (HRF2) | 48.7 (1x5) | 20.7 (1x10) | 12.7 (1x20) | 9.29 (1x50) | 8.28 (2x50) | 7.08 (2x250) |
> >
> > > To ensure consistency across the paper, for both the baseline and our approach, we followed our CIFAR-10 architecture setup described in Appendix F.2, but used attention resolution “16,8” as opposed to just “16”, decreased the learning rate (to 1e-4 from 2e-4), and increased the batch size (to 512 from 128).
> >
> > > We think those FID values are reasonable compared to the ones reported by Lipman et al. (2023), as our architecture differs in depth (2 vs. their 3), channel size (128 vs. their 256), batch size (512 vs. their 1024), and learning rate scheduler (fixed vs. their polynomial decay).

---

> > > ### Comment · Reviewer_4Etx · 2024-12-03
> > >
> > > Thank you for your response and for conducting the additional experiments. The current ImageNet results are promising, and they have increased my confidence in the official score I have assigned to the paper.

---

### Official Review · Reviewer_Cauw · 2024-11-02

**Soundness:** 2
**Presentation:** 3
**Contribution:** 2
**Rating:** 6
**Confidence:** 3

**Summary:**

The work enriches the concept of rectified flow for generating data from known base distribution to data distribution. Compared to the base approach, hierarchical rectified flow formulation models the multi-modal random velocity field and acceleration field, which leads to generating more interesting trajectories. The authors evaluate their approach on some low-dimensional use cases and small image benchmark data.

**Strengths:**

***Clarity and Accessibility***: The paper is well-written and presented in a logical manner, making the concepts and methodology easy to follow. Complex ideas are explained in a way that is accessible to readers with varying levels of familiarity with flow-based generative models, facilitating broader understanding and engagement with the research.

***Clear and Impactful Motivation***: The authors provide a compelling motivation for developing hierarchical rectified flow models, addressing limitations in existing flow-based generative models. This work has the potential to significantly impact the field by introducing a more flexible and expressive approach, which could inspire further research and applications in generative modeling.

***Solid Theoretical Foundations***: The paper presents strong theoretical foundations that support the proposed hierarchical model. These well-motivated considerations provide a rigorous basis for understanding how the model improves upon previous approaches, making the proposed framework robust and trustworthy.

***Intuitive Experimental Demonstrations***: The inclusion of low-dimensional toy experiments adds an important educational aspect to the study, offering readers an intuitive way to grasp the dynamics of the approach. These experiments clarify how the model handles multi-modal distributions and complex trajectories, offering insights that make the approach more transparent and easier to analyze.

**Weaknesses:**

***Ambiguity in Extended Hierarchical Approach***: Although the acceleration-based approach and its motivation are clear and well-justified, the transition to the extended hierarchical flow model lacks clarity. Specifically, while the training objective for the acceleration-based approach is defined by Equation (8), the relationship to the hierarchical model’s training objective, outlined in Equation (10), is not thoroughly explained. The conceptual progression and the structural specifics of the higher-dimensional source distribution (D-dimensional) are left somewhat ambiguous. Additional details on how this source distribution is constructed and how it interfaces with the hierarchical model would clarify the extension.

***Scalability and Computational Complexity***: While the hierarchical rectified flow method is innovative, its computational demands are significant, particularly when scaling to high-dimensional data. Even when considering latent models, the approach appears to be resource-intensive and potentially impractical for high-dimensional images due to its complex sampling procedure. This aspect could hinder its use in real-world scenarios where efficient sampling and rapid training times are crucial. The authors should provide a more detailed analysis of the computational overhead, comparing training and sampling times with other flow-based generative models to highlight both the benefits and trade-offs of their approach.

***Limited Scope of Experiments***: The experimental evaluation primarily uses small, low-dimensional image datasets, which limits the generalizability and relevance of the results. Given current advancements in generative modeling, such datasets do not fully showcase the potential of the hierarchical approach. Adding experiments on higher-dimensional or more complex data—perhaps by incorporating latent diffusion architectures—would better demonstrate the method’s scalability and practical value. More comprehensive experimentation would help readers assess how well the model performs in scenarios closer to those encountered in modern applications of generative models.

**Questions:**

Please refer to the weakness section.

---

> ### Author Response · Authors · 2024-11-20
> **Response to Reviewer Cauw**
>
> Thanks a lot for your time and feedback and for assessing the paper as having a clear and impactful motivation, a solid theoretical foundation, and intuitive experimental demonstrations.
>
> ***QB1: Ambiguity in Extended Hierarchical Approach***
> > The paragraph below Eq. (10) explains the definition of each vector. In the newly added Appendix E we present a more detailed derivation in order to clarify further.
>
> ***QB2: Scalability and Computational Complexity***
> > For synthetic data, we included details regarding the training/inference time in the newly added Appendix G (see Table 2 and Table 3). For MNIST and CIFAR-10, we compared training/inference time and generation performance of the baseline RF and of our method in the newly added Appendix H.3 of the revised appendix (see Tables 4-6). We briefly summarize the results here. For MNIST, our model outperforms the baseline while maintaining a comparable model size, training time, and inference time. For CIFAR-10, although our model is 1.25x larger and has a roughly 1.4x slower inference time, it still achieves superior performance compared to the baseline: 3.713 FID vs. 3.927 FID for the baseline. We think this trade-off is justified.
>
>
> ***QB3: Scope of Experiments***
> > We added additional experiments in Appendix H.2 and Appendix H.3, demonstrating compelling results on synthetic data, MNIST, and CIFAR-10. Unfortunately we don’t have the computational resources to conduct experiments on higher-dimensional data like ImageNet. However, our core idea---the hierarchical structure---is theoretically applicable to any (latent) diffusion architecture that utilizes flow matching for learning a vector field. We are open to exploring such experiments in future work and are very excited to collaborate with any interested parties.

---

> > ### Author Response · Authors · 2024-11-26
> > **Response to Reviewer Cauw**
> >
> > Thanks a lot again for your time and valuable feedback. We hope that our response and the updated paper answered your questions. Please let us know about any further questions or comments that remain. Thank you once again, and we look forward to your reply.

---

> > > ### Comment · Reviewer_Cauw · 2024-11-26
> > >
> > > Dear authors, thank you for clarifying the eq. (10)  and incorporating additional results for higher-dimensional data. In my opinion, the application of the proposed model in latent space for high-dimensional image data would make the approach more practical and highlight the benefits. However, the current, revised version is good enough for publication, therefore I decided to increase the score.

---

### Official Review · Reviewer_tEsT · 2024-11-03

**Soundness:** 2
**Presentation:** 3
**Contribution:** 2
**Rating:** 6
**Confidence:** 3

**Summary:**

This paper introduces a novel framework, hierarchical rectified flow, to model data distributions. It addresses the limitations of conventional rectified flow (RF), which only captures mean velocity, by incorporating distributional information in the velocity space. The objective, equivalent to acceleration matching, derives target acceleration from the data and prior velocity distributions. The framework can be extended beyond acceleration, creating a hierarchical generation model. The proposed approach offers benefits such as easier path integration and improved results with fewer NFEs.

**Strengths:**

The strengths of the paper are listed below:
- Clear motivation with well-written, easy-to-follow presentation.
- Experimental results that partially support the theoretical claims.

**Weaknesses:**

The weaknesses of the paper are listed below:
- While the paper's motivation is sound, my main concern lies in the practicality and application of the proposed approach. The generation framework demands significantly more model calls compared to the conventional RF framework, specifically NFEs multiplied by the number of discretizations in the velocity space. Although RF only targets mean velocity, it delivers strong empirical results with far greater efficiency than the proposed method. Additionally, RF can be rectified multiple times for better results. I recommend that the authors conduct a comprehensive comparison between the rectified RF and the proposed method. This inefficiency could lead to significantly limited applicability, especially for larger datasets like ImageNet. Could the authors include a comparison of generation performance and training/inference time between the rectified RF and the proposed method on the existing datasets, or ideally on ImageNet?

- The inefficiency becomes even more pronounced with increased depth; for example, a D-depth model requires a substantially larger architecture compared to a one-depth model.  Does the forgetting problem of the optimal trajectory occur with increased depth, necessitating an expanded architecture? Could the authors provide an ablation study on how model performance and computational requirements change as depth increases?

- The paper lacks a comprehensive comparison of efficiency, including training and inference time, against RF and rectified RF. The paper should include details of the neural network architecture to ensure reproducibility in toy settings. The paper should include more details of the neural network architecture and experimental settings to ensure reproducibility in toy scenarios.

- On the theoretical side, does increasing depth make it harder for the model to converge during training? It would be helpful to include empirical evidence of convergence rates for models with different depths or a discussion of any theoretical bounds on convergence that might exist when using larger depths.

- Minor typo on line 156: it should be \pi_0. Please review and correct all typos.

Note: The review has been edited to make it more actionable for the authors, as suggested by the PCs.

**Questions:**

Please see the Weaknesses section for my concerns and questions.

---

> ### Author Response · Authors · 2024-11-20
> **Response to Reviewer tEsT**
>
> Thanks a lot for your time and feedback and for assessing the paper as being well written.
>
> ***QA1: The generation framework demands significantly more model calls compared to the conventional RF framework, specifically NFEs multiplied by the number of discretizations in the velocity space***
> > ***We think there is a misunderstanding as our model does __not__ require significantly more neural function evaluations (NFEs).*** Note, all NFEs in our paper are total NFEs, i.e.,  $\text{NFE}=\prod_d N^{(d)}$, where $N^{(d)}$ refers to the integration steps at depth $d$. For instance, in Figure 4(b), the “HRF2 20 v steps” line at NFE = 200 uses 10 x-steps and 20 $v$-steps ($10 \times 20 = 200$). In contrast, the baseline RF line at NFE = 200 uses 200 $x$-steps. This ensures that the comparison is fair and that ***our framework does not require more NFEs***. To avoid this misunderstanding, we added L349 in the revised paper. Further, all figure axis labels have been updated to “Total NFEs”. To clarify further, we added an ablation study in Appendix G, analyzing NFEs and the impact of integration steps across depths.
>
> ***QA2: Comparison with rectified RF.***
> > Approaches for straightening the paths in flow matching models are orthogonal to our work. In the original paper, we reviewed those methods at the end of Section 5 and mentioned that they can be adopted in our formulation. To demonstrate this, we have added Appendix H.2 which incorporates OTCFM in our framework. We find that HRF further improves OTCFM. We use OTCFM here since it’s a single-step training approach to straighten the paths.
>
> ***QA3: Training/Inference time comparison***
> > For synthetic data, we included details regarding the training/inference time in the newly added Appendix G (see Table 2 and Table 3). For MNIST and CIFAR-10, we compared training/inference time and generation performance of the baseline RF and of our method in the newly added Appendix H.3 of the revised appendix (see Tables 4-6). We briefly summarize the results here. For MNIST, our model outperforms the baseline while maintaining a comparable model size, training time, and inference time. For CIFAR-10, although our model is 1.25x larger and has a roughly 1.4x slower inference time, it still achieves superior performance compared to the baseline: 3.713 FID vs. 3.927 FID for the baseline. We think this trade-off is justified.
>
> ***QA4: ImageNet results***
> > Unfortunately we don’t have the computational resources to conduct experiments on higher-dimensional data like ImageNet. However, our core idea---the hierarchical structure---is theoretically applicable to any (latent) diffusion architecture that utilizes flow matching for learning a vector field. We are open to exploring such experiments in future work and are very excited to collaborate with any interested parties.
>
>
> ***QA5: Could the authors provide an ablation study on how model performance and computational requirements change as depth increases?***
> > We included ablation studies in the revised Appendix G. The computational requirements and performance of HRF are mostly independent of the depth, but rather depend on the number of chosen integration/sampling steps. In general, higher depth HRF gives better performance with an acceptable increase in model size, training time, and inference time. For synthetic data please check the newly added Table 2 and Table 3 for details. For MNIST and CIFAR-10, please see the newly added Tables 4-6.
>
> ***QA6: Network architecture and reproducibility***
> > The network architectures are described in Appendix F.1 and F.2. Note, we revised Appendix F.2 to improve clarity and added Table 2 and Table 4 to compare architectures. We will also release the code with complete hyperparameter settings to ensure full reproducibility of our synthetic examples and experiments on MNIST and CIFAR-10, as stated in L023.
>
> ***QA7: Training Convergence***
> > We added Figure 8 in Appendix G to show that training of models with different depths remains stable and convergence behavior doesn’t differ significantly from the baseline.
>
> ***QA8: Minor typo on line 156: it should be \pi_0. Please review and correct typos.***
> > There is no typo in line 156. $\pi_1$ is the velocity distribution at $\tau = 1$, while $t$ is the time variable for the data domain. To clarify we have changed the notation for the velocity distribution $\pi_1(x_t, t)$ to $\pi_1(v; x_t, t)$ throughout the paper. We have also checked the paper again and corrected typos.

---

> > ### Comment · Reviewer_tEsT · 2024-11-25
> > **Comments by Reviewer tEsT**
> >
> > Thank you for your response, which has primarily addressed my concerns. I have additional additional questions and concerns based on your revised paper before reaching a final decision:
> >
> > - In table 6, why the performance of MNIST using 100 and 500 NFEs is worse than using 50 NFEs?
> > - Could you please provide a justification for why using greater depths results in the lowest loss curve compared to shallower depths and RF? Would this trend likely continue with HRF4 and beyond?

---

> > > ### Author Response · Authors · 2024-11-26
> > > **Response to Reviewer tEsT**
> > >
> > > Thanks a lot for your timely reply and your valuable suggestions.
> > >
> > > ***QA9: In Table 6, why the performance of MNIST using 100 and 500 NFEs is worse than using 50 NFEs?***
> > > > Thanks for pointing this out. We revised Table 7 (previous Table 6) by repeating all the FID computations 10 times to provide standard deviations. Moreover, for 100 and 500 NFEs we adjusted the sampling schedule from (5,20) and (5,100) to (10,10) and (100,5). We no longer observe FIDs to get worse when increasing the NFEs.
> > >
> > > ***QA10: Could you please provide a justification for why using greater depths results in the lowest loss curve compared to shallower depths and RF? Would this trend likely continue with HRF4 and beyond?***
> > > > We revised Figure 8 to include HRF4 and HRF5 results. We also added the following discussion to the revised appendix: “Importantly, note that Figure 8 mainly serves to compare convergence behavior and not loss magnitudes as those magnitudes reflect different objects, i.e., velocity for a depth of 1, acceleration for a depth of 2, etc. Moreover, the deep net structure for the functional field of directions $f$ depends on the depth, which makes a direct comparison of the loss more challenging.”

---

> ### Comment · Reviewer_tEsT · 2024-11-26
> **Comments by Reviewer tEsT (part 2)**
>
> Thank you for your response. I appreciate the authors' efforts in revising the paper and addressing the concerns raised. I find the theoretical insights to be interesting and believe they hold potential for meaningful contributions to the field. While the paper may have certain practical limitations, particularly when applied to large datasets with greater depths, the performance of the 2-depth version is comparably strong without requiring excessive resources. Based on this, I will raise my score to 6.

---

### Author Response · Authors · 2024-11-20
**Summary of Changes**

We thank all reviewers for their time and feedback. We are excited to see that our work was assessed as well-written (R1, R2, R3), with clear and impactful motivation, solid theoretical foundation, intuitive experimental demonstration (R2), and interesting theorems (R3). We updated the paper and appendix to answer reviewer questions. Additions include: 1) the density estimation in Appendix D, 2) more details on the relationship between the objectives given in Eq. (8) and Eq. (10) in Appendix E, 3) details on the models in Appendix F, 4) ablation studies in Appendix G, 5) adopting minibatch optimal transport into HRF in Appendix H.2, and 6) additional results on MNIST and CIFAR-10 in Appendix H.3. We also clarified that **our comparison is fair** because the NFEs reported in the paper are the total NFEs, i.e., the product of the number of integration steps at all HRF levels (L349 of the revised paper and updated figure axis). We answer questions for each reviewer individually and point to the corresponding paper and appendix additions.

---

### Meta-Review · Area_Chair_UCNN · 2024-12-19

**Metareview:**

The paper proposes the Hierarchical Rectified Flow that allows to model multiple ODEs instead of a single ODE in the regular Rectified Flow. It allows to better represent multi-modal distributions, e.g. for physical simulations. The method also allows to reduce the number of neural function evaluations and considerably speed up the forward pass. The paper has solid theoretical foundations, supported by experiments. Among weaknesses, the experiments primarily use small, low-dimensional image datasets, and it is not clear how the model will scale to larger modalities.

**Additional Comments On Reviewer Discussion:**

The authors have expanded the appendix and added more method details. Reviewers had concerns about fair comparison to rectified flow and the computational cost, which the authors clarified in the rebuttal

---

### Decision · Program_Chairs · 2025-01-22

Accept (Poster)